# REDUCR: Robust Data Downsampling using Class Priority Reweighting

**William Bankes**
Department of Computer Science
University College London
william.bankes.21@ucl.ac.uk

**George Hughes**
Department of Computer Science
University College London

**Ilija Bogunovic**∗
Department of Electrical Engineering
University College London
i.bogunovic@ucl.ac.uk

**Zi Wang**∗
Google DeepMind
wangzi@google.com

## Abstract

Modern machine learning models are becoming increasingly expensive to train for real-world image and text classification tasks, where massive web-scale data is collected in a streaming fashion. To reduce the training cost, online batch selection techniques have been developed to choose the most informative datapoints. However, many existing techniques are not robust to class imbalance and distributional shifts, and can suffer from poor worst-class generalization performance. This work introduces REDUCR, a robust and efficient data downsampling method that uses class priority reweighting. REDUCR *reduces* the training data while preserving worst-class generalization performance. REDUCR assigns priority weights to datapoints in a class-aware manner using an online learning algorithm. We demonstrate the data efficiency and robust performance of REDUCR on vision and text classification tasks. On web-scraped datasets with imbalanced class distributions, REDUCR significantly improves worst-class test accuracy (and average accuracy), surpassing state-of-the-art methods by around 15%.

## 1 Introduction

The abundance of data has had a profound impact on machine learning (ML), both positive and negative. On the one hand, it has enabled ML models to achieve unprecedented performance on a wide range of tasks, such as image and text classification [Kuznetsova et al., 2020, He et al., 2015, Brown et al., 2020, Tran et al., 2022, Anil et al., 2023]. On the other hand, training models on such large datasets can demand significant computational resources [Kaddour et al., 2023], making it unsustainable in some situations [Bender et al., 2021, Patterson et al., 2021]. Additionally, the high speed at which streaming data is collected can make it infeasible to train on all of the data before deployment. To tackle these issues, various methods have emerged to selectively choose training data, either through pre-training data pruning [Sorscher et al., 2022, Bachem et al., 2017] or online batch selection techniques [Loshchilov and Hutter, 2016, Mindermann et al., 2022], ultimately reducing data requirements and enabling ML models to handle otherwise unmanageable large and complex datasets.

In real-world settings, a variety of factors can affect the selection of datapoints, such as noise [Xiao et al., 2015, Cao et al., 2021, Wei et al., 2021] and class-imbalance in the data [Van Horn et al., 2018, Philip and Chan, 1998, Radivojac et al., 2004]. Online selection methods can exacerbate

---

∗Co-senior authors. Code available at: https://github.com/williambankes/REDUCR.

38th Conference on Neural Information Processing Systems (NeurIPS 2024).

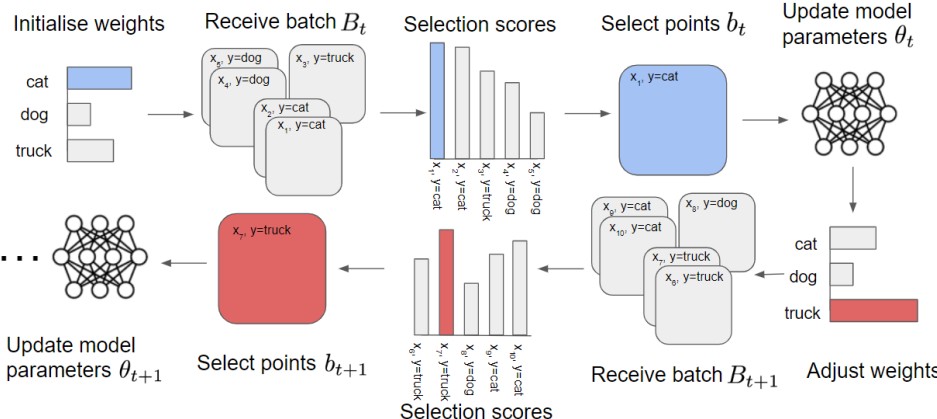

Figure 1: REDUCR starts by initializing weights of classes. At each timestep $t$, the model receives a batch of datapoints $B_t$. REDUCR computes the selection scores for each datapoint based on its usefulness to the model and the class weights, and selects new datapoints $b_t \subset B_t$ that achieve the highest selection scores. After the model takes gradient steps on the selected datapoints, REDUCR adjusts the weights to reflect increased priorities on underperforming classes.

these problems by further reducing the number of datapoints from underrepresented classes, which can degrade the performance of the model on those classes [Buda et al., 2018, Cui et al., 2019]. Moreover, distributional shift [Koh et al., 2021] between training and test time can lead to increased generalization error if classes with poor generalization error are overrepresented at test time.

In this work, we introduce REDUCR, which is a new online batch selection method that is *robust* to noise, imbalance, and distributional shifts. REDUCR employs multiplicative weights update to reweight and prioritize classes that are performing poorly during online batch selection. Figure 1 illustrates the intuition behind how the method works. REDUCR can effectively reduce the training data while preserving the worst-class generalization performance of the model. For example, on the Clothing1M dataset [Xiao et al., 2015], Figure 2 shows that, compared to the best performing online batch selection methods, REDUCR achieves around a 15% boost in performance for the worst-class test accuracy.

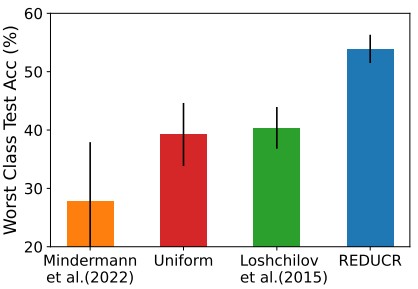

Figure 2: REDUCR significantly improves worst-class test accuracy on Clothing1M outperforming Uniform and other recent works.

**Main contributions.** (1) We formalise the maximin problem of robust data downsampling (§3). (2) We propose the REDUCR algorithm, which is equipped with a new robust selection rule that evaluates how much datapoints will affect the generalization error of a specific class (§4.2). (3) We evaluate our algorithm on a series of text and image classification tasks and show that it achieves strong worst-class test accuracy while frequently surpassing state-of-the-art methods in terms of average test accuracy(§5).

**Related work.** Mindermann et al. [2022] have developed an online batch selection method called RHO-LOSS, which uses a *reference model* trained on a holdout dataset to guide the selection of points during training. Certain extensions of this work have focused on using a reference model in different settings such as reinforcement learning [Sujit et al., 2022]. However, to our knowledge, none have focused on improving the worst-class generalisation performance. Other batch selection methods [Loshchilov and Hutter, 2016, Jiang et al., 2019, Kawaguchi and Lu, 2020] use the training loss of points under the model or an approximate gradient norm [Katharopoulos and Fleuret, 2017] to select challenging points. We observe that these methods (e.g., see Loshchilov and Hutter [2016] in Figure 2) exhibit greater consistency in terms of worst-class generalization error in imbalanced datasets. Nevertheless, Loshchilov and Hutter [2016] do not surpass the average generalization error achieved by point selection with a reference model, namely, RHO-LOSS. Recently, several works have also used reference models or a holdout dataset to train robust models. Oren et al. [2019], Liu et al. [2021], Clark et al. [2019] use a reference model to identify difficult-to-learn groups (or points, or biases) during

training. Han et al. [2018] use two models which act as a reference model for the other to remove noisy points from the training data. Cao et al. [2021], Ren et al. [2018] use a holdout dataset to reweight points or their regularization during training to achieve the best loss on the validation holdout dataset.

Sagawa et al. [2020] reweight groups known at training time and focus on fighting spurious correlations and improving worst-group generalisation error. In contrast, in our setting, class labels are available and we measure the performance in terms of worst-class generalisation error. Moreover, whilst these works aim to train robust models they do not consider efficient data downsampling strategies. The approach of Sagawa et al. [2020] to group robustness considers a small number of groups (up to 4 in their empirical study). Similarly, in our work, we consider classification settings with a controlled number of classes ($< 1000$) as the problem of robustness becomes less applicable in settings where the number of classes are high.

Xie et al. [2023] use both weights update rules and a reference model to find mixtures of corpora in LLM pretraining resulting in improved performance and training speed. Besides the problem setup, our method differs in three ways: i) we focus upon online batch selection; ii) we use multiple reference models; iii) and we use a class-holdout loss term (see Equation (8)) to reweight batches. Efficient data downsampling is a well-explored problem with various approaches, including active learning methods when label information is unknown [MacKay, 1992, Houlsby et al., 2011, Kirsch et al., 2019, 2023, Ash et al., 2020]; data pruning and coreset techniques for pre-training data downsampling [Sorscher et al., 2022, Bachem et al., 2017, Borsos et al., 2020, Coleman et al., 2020]; data distillation approaches [Cazenavette et al., 2022, Nguyen et al., 2021]; and non-parametric inducing point methods [Galy-Fajou and Opper, 2021].

## 2 Background

We consider a $C$-way classification task and denote a model as $p(y \mid x, \theta)$, where $x$ denotes an input and $y \in [C]$ the corresponding class label; the model is parameterized by $\theta$. For any training dataset $\mathcal{D} = \{(x_i, y_i)\}_{i=1}^{N}$ with $N$ datapoints, we use a point estimate of $\theta$ to approximate the posterior model as $p(y \mid x, \mathcal{D}) \approx p(y \mid x, \hat{\theta})$. This estimate $\hat{\theta}$ can be obtained by running stochastic gradient descent (SGD) to optimize the cross-entropy loss over a training dataset $\mathcal{D}$.

The goal of *data downsampling* is to select a dataset $\mathcal{D}_T \subset \mathcal{D}$ of size $T$ ($\ll N$) for training such that the generalisation error of the resulting model is minimised. We write this objective in terms of a separate *holdout* dataset $\mathcal{D}_{ho} = \{(x_{ho,i}, y_{ho,i})\}_{i=1}^{N_{ho}}$ as follows:

$$\mathcal{D}_T = \underset{D \subset \mathcal{D}, |D|=T}{\operatorname{argmax}} \ \log p(\mathbf{y}_{ho}|\mathbf{x}_{ho}, D), \tag{1}$$

where the inputs and their labels are $\mathbf{x}_{ho} = [x_{i,ho}]_{i=1}^{N_{ho}}$ and $\mathbf{y}_{ho} = [y_{i,ho}]_{i=1}^{N_{ho}}$, respectively. Here, the likelihood of the holdout dataset is used as a proxy for the generalisation error. The problem is computationally prohibitive due to its combinatorial nature. Moreover, for a massive (or streaming) training dataset $\mathcal{D}$, it is not computationally possible to load $\mathcal{D}$ all at once and it is common to loop through the data by iteratively loading subsets.

**Online batch selection** is a practical streaming setup to approximate the data downsampling problem, where at each timestep $t$, the model observes a training data subset $B_t \subset \mathcal{D}$, and the goal is to iteratively select a small batch $b_t \subset B_t$ for the model to take gradient steps. A standard solution to this problem is to design a selection score function that take into account the labels of the data. The selection score function can then be used to *score* the utility of the small batch $b_t$. See Algorithm 2 in Appendix A.1 for an example method.

**Reducible Holdout Loss (RHO-Loss)** [Mindermann et al., 2022] is an online batch selection method that uses the performance on a holdout dataset as the selection scores for small batches. More precisely, for each timestep $t$, RHO-Loss selects

$$b_t = \underset{b \subset B_t}{\operatorname{argmax}} \ \log p(\mathbf{y}_{ho} \mid \mathbf{x}_{ho}, \mathcal{D}_t \cup b), \tag{2}$$

where $\mathcal{D}_t = \bigcup_{\tau=1}^{t-1} b_\tau$ is the cumulative training data the model has encountered until iteration $t$.

## 3 Problem Formulation

In this work, we introduce the *robust* data downsampling problem, where the goal is to select a training dataset $\mathcal{D}_T$ of size $T$ such that worst-class performance is optimized. Let the holdout dataset with class $c \in [C]$ be $D_{ho}^{(c)} = \{(x, y) \in D_{ho} \mid y \equiv c\} = \{(x_{ho,i}^{(c)}, y_{ho,i}^{(c)})\}_{i=1}^{N_{ho}^{(c)}}$. We can write the objective of robust data downsampling as

$$\mathcal{D}_T = \underset{D \subset \mathcal{D}, |D|=T}{\operatorname{argmax}} \min_{c \in [C]} \log p(\mathbf{y}_{ho}^{(c)} \mid \mathbf{x}_{ho}^{(c)}, D), \tag{3}$$

where $\mathbf{x}_{ho}^{(c)} = [x_{ho,i}^{(c)}]_{i=1}^{N_{ho}^{(c)}}$ and $\mathbf{y}_{ho}^{(c)} = [y_{ho,i}^{(c)}]_{i=1}^{N_{ho}^{(c)}}$ correspond to the collections of inputs and labels in the class-specific holdout dataset $D_{ho}^{(c)}$.

Compared to Equation (1), the objective in Equation (3) is even more challenging because of the maximin optimisation that involves $C$ discrete classes. In fact, solving Equation (3) is known to be NP-hard, even when the objectives (each $p(\mathbf{y}_{ho}^{(c)}|\mathbf{x}_{ho}^{(c)}, \cdot)$, $c \in [C]$) are *submodular* set functions. Chen et al. [2017] demonstrate the application of zero-sum game no-regret dynamics, where a learner employs a $(1 - 1/e)$-near-optimal greedy strategy and an adversary seeks to find a distribution over loss functions that maximizes the learner's loss. In this scenario, a single set is identified, which, although larger than size $T$, achieves a constant-factor approximation.

**Robust online batch selection** approximates the robust data downsampling problem by taking into account the practical limitations of data operation. Namely, we assume a streaming setting where the model observes training data subset $B_t \subset \mathcal{D}$ at each timestep $t$. The goal is to select a small batch $b_t \subset B_t$ to compute gradients for model training with SGD, such that the model obtains top performance for the worst-class (Equation (3)). The robust setting motivates the development of novel batch selection methods that consider how each datapoint affects the generalization error on the worst-case class of inputs, rather than just the overall generalization error. Next, we introduce a new selection rule that achieves this and propose a practical algorithm for its implementation.

## 4 REDUCR for Robust Online Batch Selection

We propose REDUCR, a robust and efficient data downsampling method using class priority reweighting to solve the robust online batch selection problem in Section 3. The batch selection strategy of REDUCR relates the effect of training on a batch of candidate points $b_t$ to the generalization error of a specific class in the holdout dataset.

### 4.1 Online Learning

To solve Equation (3) in an online manner, we propose to use class priority reweighting, a variant of the multiplicative weights update method [Freund and Schapire, 1997, Cesa-Bianchi and Lugosi, 2006, Sessa et al., 2019]. At the beginning of training we initialise a weight vector $\mathbf{w}_0$ over a $C$ dimensional simplex, $\Delta = \{\mathbf{w} = [w_c]_{c=1}^C \in \mathbb{R}^C | \sum_{c=1}^C w_c = 1\}$. Each element of $\mathbf{w}_0$ is initialised to be $w_{0,c} = 1/C$. For each iteration $t$, small batch $b_t \subset B_t$ is chosen by maximising the weighted sum of the $C$ different class-specific scoring functions (i.e., by best-responding to the current class-weights $\mathbf{w}_t$),

$$b_t = \underset{b \subset B_t}{\operatorname{argmax}} \sum_{c=1}^C w_{t,c} \left( \log p(\mathbf{y}_{ho}^{(c)}|\mathbf{x}_{ho}^{(c)}, \mathcal{D}_t \cup b) \right), \tag{4}$$

where $\mathcal{D}_t = \bigcup_{\tau=1}^{t-1} b_\tau$, $\mathbf{w}_t = [w_{t,c}]_{c=1}^C \in \Delta$, and

$$w_{t,c} = w_{t-1,c} \frac{\exp\left(-\eta \log p(\mathbf{y}_{ho}^{(c)}|\mathbf{x}_{ho}^{(c)}, \mathcal{D}_t)\right)}{\sum_{j=1}^C w_{t-1,j} \exp\left(-\eta \log p(\mathbf{y}_{ho}^{(j)}|\mathbf{x}_{ho}^{(j)}, \mathcal{D}_t)\right)}. \tag{5}$$

In the previous alternating procedure, class-weights are updated multiplicatively according to how well they perform given the selected batch, they increase for poorly performing classes and decrease otherwise. In Equation (5), $\eta$ is a learning rate that adjusts how concentrated the probability mass is in the resulting distribution. Figure 1 shows an intuitive illustration of how reweighting works in practice where classes that perform badly have low data likelihoods and are thus upweighted by

Equation (5). In Appendix A.7.3 we explore an alternative solution to solve Equation (3); we solve an approximate robust optimisation problem directly at every timestep $t$ and empirically demonstrate the multiplicative weights method outperforms it. We next introduce how to compute the likelihoods for class-specific holdout sets, i.e., $p(\mathbf{y}_{ho}^{(c)}|\mathbf{x}_{ho}^{(c)}, \mathcal{D}_t \cup b)$ in Equation (4).

## 4.2 Computing selection scores

Given the current dataset $\mathcal{D}_t$ at timestep $t$ and additional datapoints $b \subset B_t$, we would like to compute the likelihood of the holdout dataset that belongs to class $c$. For simplicity, we consider the case where the small batch to be selected only includes a single datapoint, i.e., $b = \{(x, y)\}$. We express the objective using a Bayesian perspective,

$$\log p(\mathbf{y}_{ho}^{(c)} \mid \mathbf{x}_{ho}^{(c)}, \mathcal{D}_t \cup \{(x,y)\}) = \log \frac{p(y|x, \mathcal{D}_{ho}^{(c)}, \mathcal{D}_t) p(\mathbf{y}_{ho}^{(c)}|\mathbf{x}_{ho}^{(c)}, x, \mathcal{D}_t)}{p(y|x, \mathbf{x}_{ho}^{(c)}, \mathcal{D}_t)} \quad (6)$$

$$= \log \frac{p(y|x, \mathcal{D}_{ho}^{(c)}, \mathcal{D}_t) p(\mathbf{y}_{ho}^{(c)}|\mathbf{x}_{ho}^{(c)}, \mathcal{D}_t)}{p(y|x, \mathcal{D}_t)} \quad (7)$$

$$= -\log p(y \mid x, \mathcal{D}_t) + \log p(y \mid x, \mathcal{D}_t, \mathcal{D}_{ho}^{(c)}) + \log p(\mathbf{y}_{ho}^{(c)} \mid \mathbf{x}_{ho}^{(c)}, \mathcal{D}_t).$$

Equation (6) follows from the application of the Bayes rule and the conditional independence of $x$ and $\mathbf{x}_{ho}^{(c)}$ with $\mathbf{y}_{ho}^{(c)}$ and $y$, respectively. The posterior terms in Equation (6) can be approximated with point estimates of model parameters (see §2). Computing Equation (6) involves two models: (1) the *target* model with parameters $\theta_t$, which is trained on the cumulative training dataset $\mathcal{D}_t = \bigcup_{\tau=1}^{t-1} b_\tau$; (2) a *class-irreducible loss model* (following the terminology from Mindermann et al. [2022]) with parameters $\theta_t^{(c)}$, which is trained on $\mathcal{D}_t$ and class-specific holdout data $\mathcal{D}_{ho}^{(c)}$. The target model is what we are interested in for the classification task. We use $\mathcal{L}[y|x, \theta] = -\log p(y \mid x, \theta)$ to denote the cross-entropy loss for any model parameters $\theta$, and we re-write Equation (6) as follows,

$$\log p(\mathbf{y}_{ho}^{(c)} \mid \mathbf{x}_{ho}^{(c)}, \mathcal{D}_t \cup \{(x,y)\}) \approx \underbrace{\mathcal{L}[y|x, \theta_t]}_{\text{model loss}} - \underbrace{\mathcal{L}[y|x, \theta_t^{(c)}]}_{\text{class-irreducible loss}} - \underbrace{\mathcal{L}[\mathbf{y}_{ho}^{(c)}|\mathbf{x}_{ho}^{(c)}, \theta_t]}_{\text{class-holdout loss}}. \quad (8)$$

We name the three terms in Equation (8) the *model loss*, *class-irreducible loss* and *class-holdout loss*, respectively. We define the term *excess loss* as the difference of the model loss and class-irreducible loss. The excess loss is the improvement in loss for point $(x, y)$ by observing more data from class $c$ (i.e., $\mathcal{D}_{ho}^{(c)}$). Intuitively, if two data points are from different classes, REDUCR will take into account the weight of the worst-performing class, which is reflected by the class-holdout loss. This ensures that REDUCR is focusing on improving the performance of the model on the classes that are most difficult to learn. In a different scenario, if two datapoints are from the same class, their class-holdout losses will be the same, and the point with a larger excess loss will be preferred. This means that REDUCR prefers datapoints whose losses have more potential to be improved.

Computing the approximate in Equation (8) is far more tractable than naively re-training a new model (i.e., $\log p(\mathbf{y}_{ho}^{(c)}|\mathbf{x}_{ho}^{(c)}, \mathcal{D}_t \cup \{(x,y)\})$) for each possible candidate point $(x, y)$. The model loss and the class-holdout loss only require evaluating the cross-entropy losses of some datapoints on the target model. More generaly, if batch $b$ can include more than one point, we can simply change the $x$ and $y$ to a list of inputs and labels instead. Next, we further improve the efficiency of REDUCR by approximating the class-irreducible loss model.

## 4.3 Class-Irreducible Loss Models

For each selected batch $b_t$ under the current selection rule in Equation (8), we need to update $C$ class-irreducible loss models to compute the class-irreducible losses. We propose to approximate these models using *amortised* class-irreducible loss models, which are trained for each class at the beginning of REDUCR and do not need to be updated during online batch selection.

We interpret the class irreducible loss term as an expert model at predicting the label of points from a specific class $c$ due to the extra data from the holdout dataset this term has available. To create an approximation of this expert model, we train the amortised class-irreducible loss models using an adjusted loss function in which points with a label from the class $c$ are up-weighted by a parameter $\gamma \in (0, +\infty)$ (set in Section 5):

$$\phi_c = \arg\min_\phi \sum_{(x,y) \in \mathcal{D}_{\wr}} (1 + \gamma \, \mathbb{I}[c \equiv y]) \, \mathcal{L}[y|x, \phi]. \quad (9)$$

---

**Algorithm 1** REDUCR for robust online batch selection

---

1: **Input:** data pool $\mathcal{D}$, holdout data $\mathcal{D}_{ho} = \bigcup_{c \in C} \mathcal{D}_{ho}^{(c)}$, learning rate $\eta \in (0, \infty)$, small batch size $k$, total timesteps $T/k$
2: Initialize class weights $\mathbf{w}_1 = \frac{1}{C}\mathbf{1}_C$
3: Use $\mathcal{D}_{ho}$ to train $C$ amortised class irreducible loss models as per Equation (9) to obtain $\phi_c$
4: **for** $t \in [T/k]$ **do**
5:     Receive batch $B_t \subset \mathcal{D}$
6:     $b_t = \underset{b \subset B_t : |b| = k}{\operatorname{argmax}} \sum_{(x,y) \in b} \sum_{c \in C} w_{t,c} \cdot \max(0, \mathcal{L}[y|x, \theta_t] - \mathcal{L}[y|x, \phi_c])$ ▷ Select points with
    top k selection scores
7:     Compute the objective value for every class $c \in C$:
    $\alpha_c = \sum_{(x,y) \in b_t} \max(0, \mathcal{L}[y|x, \theta_t] - \mathcal{L}[y|x, \phi_c]) - \mathcal{L}[\mathbf{y}_{ho}^{(c)}|\mathbf{x}_{ho}^{(c)}, \theta_t]$
8:     Update class weights for every class $c \in C$: $w_{t+1,c} = w_{t,c} \frac{\exp(-\eta \alpha_c)}{\sum_{j \in C} w_{t,j} \exp(-\eta \alpha_j)}$
9:     $\theta_{t+1} \leftarrow SGD(\theta_t, b_t)$
10: **end for**

---

Here we define $\mathbb{I}[\cdot]$ as the indicator function. Equation (9) optimizes over the parameters of the amortised class-irreducible loss model for class $c$, and obtain $\phi_c$ to approximate $\theta_t^{(c)}$ in Equation (8), i.e., $\mathcal{L}[y|x, \theta_t^{(c)}] \approx \mathcal{L}[y|x, \phi_c]$. The up-weighting of points can be considered a form of importance weighting [Shimodaira, 2000], where by up-weighting points with labels in a specific class we calculate a Monte Carlo approximation of the loss under a distribution in which points from class $c$ are more prevalent. Algorithm 3 details the full amortised class-irreducible loss model training procedure in Appendix A.2. We provide further motivation of our approximation in Appendix A.3.

## 4.4 REDUCR as a practical algorithm

We use the selection objective in Equation (8) along with the amortised class-irreducible loss model approximation (Section 4.3) and the online algorithm (Section 4.1) to reweight the worst performing class during training and select points that improve its performance. See Algorithm 1 for a full description of the REDUCR method.

At each iteration, the top $k$ points are selected (Line 6) according to the weighted sum of Equation (8) for each class $c \in C$, thus efficiently approximating the combinatorial problem from Equation (4). As the class-holdout loss does not depend on the selected points $b_t$ and we sum over the classes, we can remove this term from the weighted sum of the selection scores and only apply it when updating the weights $\mathbf{w}_t$ (in Line 7 and 8). We calculate the *average* class-holdout loss to remove any dependence of the term upon the size of the classes in the holdout dataset. We find that clipping the excess loss improves the stability of the algorithm in practice. We test this heuristic empirically in Section 5.2 and provide an intuitive explanation for why this is the case in Appendix A.7.2.

When comparing REDUCR to other online batch selection methods, we observe distinct batch selection patterns. When the dataset is class-imbalanced, the underrepresented classes tend to perform worse because of the lack of training data from those classes. RHO-LOSS may struggle to select points from the underrepresented classes as they have less effect on the loss of the holdout dataset. Selection rules that select points with high training loss [Loshchilov and Hutter, 2016, Kawaguchi and Lu, 2020, Jiang et al., 2019] might select points from the underrepresented classes but have no reference model to determine which of these points are learnable given more data and thus noisy or task-irrelevant points may be selected. In contrast, REDUCR addresses both of these issues by identifying underrepresented classes and using the class-irreducible loss model to help to determine which points from these classes should be selected.

Even when the dataset is not imbalanced, certain classes might be difficult to learn; for example, due to noise sources in the data collection processes. Via Equation (5), REDUCR is able to re-weight the selection scores such that points that are harder to learn from worse-performing classes are selected over points that are easier to learn from classes that are already performing well. This is in contrast to RHO-LOSS which will always select points that are easier to learn. We empirically demonstrate this on class balanced datasets in Section 5.

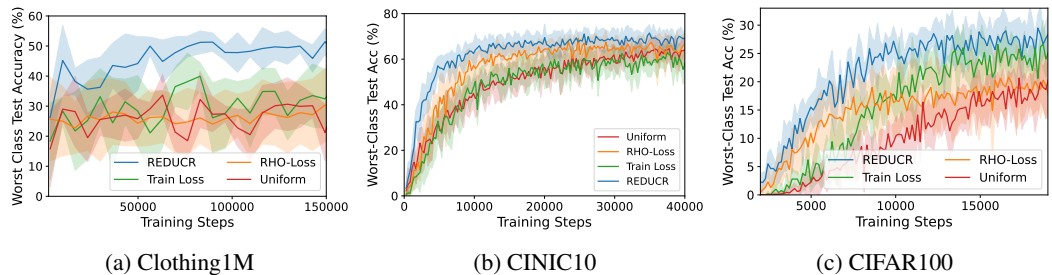

| (a) Clothing1M | (b) CINIC10 | (c) CIFAR100 |

Figure 3: REDUCR improves the worst-class test accuracy and data efficiency when compared with the RHO-LOSS, TRAIN LOSS and UNIFORM baselines on the a) Clothing1M dataset, b) the CINIC10 dataset, and c) the CIFAR100 dataset.

## 5 Experiments

In this section, we present empirical results to showcase the performance of REDUCR on large-scale vision and text classification tasks.

**Datasets.** We train and test REDUCR on image and text datasets. We use CIFAR10 [Krizhevsky et al., 2012], CINIC10 [Darlow et al., 2018], Clothing1M [Xiao et al., 2015], the Multi-Genre Natural Language Interface (MNLI), and the Quora Question Pairs (QQP) datasets from the GLUE NLP benchmark [Wang et al., 2019]. Each dataset is split into a labelled training, validation and test dataset (for details see Appendix A.5), the validation dataset is used to train the class-irreducible loss models and evaluate the class-holdout loss during training. The Clothing1M dataset uses 100k additional points from the training dataset along with the validation dataset to train the irreducible loss model(s) (as per [Mindermann et al., 2022]). We simulate the streaming setting by randomly sampling batch $B_t$ from dataset $\mathcal{D}$ at each timestep.

**Models.** For the experiments on image datasets (CIFAR10, CINIC10 and Clothing1M) all models use a ResNet-18 model architecture [He et al., 2016]. For the Clothing1M dataset we use a ResNet-18 model pretrained on the imagenet dataset [Deng et al., 2009]. The networks are optimised with AdamW [Loshchilov and Hutter, 2019] and the default Pytorch hyperparameters are used for all methods except CINIC10 for which the weight decay is set to a value of 0.1. For the NLP dataset we use the *bert-base-uncased* [Devlin et al., 2019] model from HuggingFace [Wolf et al., 2020] and set the optimizer learning rate to $1e^{-6}$.

**Baselines.** We benchmark our method against the state-of-the-art RHO-LOSS [Mindermann et al., 2022] and Loshchilov and Hutter [2016], an online batch selection method that uses the training loss to select points. We refer to the latter baseline as TRAIN LOSS. We also compare against UNIFORM where points are chosen at random from the large batch at each training step.[2] All experiments are run multiple times and the mean and standard deviation across runs calculated. Unless stated otherwise 10% of batch $B_t$ is selected as the small batch $b_t$, and we set $\eta = 1e - 4$. $\gamma = 9$ is used when training each of the amortised class-irreducible loss models on the vision datasets and $\gamma = 4$ for the NLP datasets. We study the impact of $\gamma$ and $\eta$ on REDUCR further in Appendix A.10. For full details of the experimental setup see Appendix A.5.[3]

**Metrics.** Finally, it is important to note that we analyse the worst-class test accuracy metric which can be interpreted as a lower bound on a model's performance under all class distribution shifts. This is because the worst possible distribution shift between the training and test set is one where the entire test set consists of only points from the worst performing class.

### 5.1 Key results

The worst-class and average test accuracy for the datasets and model are shown in Table 1 and Table 2, respectively. Across all datasets, REDUCR outperforms the baselines in terms of the worst-class accuracy and matches or even outperforms the average test accuracy of RHO-LOSS within one standard deviation. This is surprising because the primary goal of REDUCR is not to optimize the overall average (over classes) performance. REDUCR performs particularly strongly

---

[2]We use training step and timestep interchangeably.

[3]Code available at https://github.com/williambankes/REDUCR

Table 1: REDUCR outperforms RHO-LOSS (the best overall baseline) in terms of the worst-class test accuracy on Clothing1M, CINIC10 and CIFAR10 by at least 5-26%. Across all baselines, REDUCR gains about 15% more accuracy on the noisy and imbalanced Clothing1M dataset as shown in Figure 2. *CIFAR100 results from training step 10k where REDUCR converges, after 10k further training steps TRAIN LOSS achieves a similar performance.

| Dataset | Worst-Class Test Accuracy (%) $\pm 1$ std | | | |
|---|---|---|---|---|
| | UNIFORM | TRAIN LOSS | RHO-LOSS | REDUCR |
| CIFAR10 (10 runs) | $75.01 \pm 1.37$ | $76.1 \pm 2.31$ | $78.80 \pm 2.09$ | $\mathbf{83.29 \pm 0.84}$ |
| CINIC10 (10 runs) | $64.70 \pm 2.45$ | $64.83 \pm 4.75$ | $69.39 \pm 3.56$ | $\mathbf{75.30 \pm 0.85}$ |
| CIFAR100* (5 runs) | $10.59 \pm 3.63$ | $17.59 \pm 5.17$ | $16.0 \pm 6.93$ | $\mathbf{26.00 \pm 2.65}$ |
| Clothing1M (5 runs) | $39.23 \pm 5.41$ | $40.37 \pm 3.58$ | $27.77 \pm 10.16$ | $\mathbf{53.91 \pm 2.42}$ |
| MNLI (5 runs) | $74.70 \pm 1.26$ | $74.56 \pm 1.44$ | $76.74 \pm 0.93$ | $\mathbf{79.45 \pm 0.39}$ |
| QQP (5 runs) | $73.21 \pm 2.04$ | $79.96 \pm 2.34$ | $78.21 \pm 1.95$ | $\mathbf{86.61 \pm 0.49}$ |

Table 2: Together with Table 1, these results demonstrate that REDUCR improves the worst-class test accuracy while maintaining strong average test accuracy despite REDUCR not explicitly optimizing the average test accuracy. REDUCR outperforms the baseline methods on the CIFAR100 and Clothing1M datasets.

| Dataset | Average Test Accuracy (%) $\pm 1$ std | | | |
|---|---|---|---|---|
| | UNIFORM | TRAIN LOSS | RHO-LOSS | REDUCR |
| CIFAR10 (10 runs) | $85.09 \pm 0.52$ | $88.86 \pm 0.22$ | $\mathbf{90.00 \pm 0.33}$ | $\mathbf{90.02 \pm 0.44}$ |
| CINIC10 (10 runs) | $79.51 \pm 0.30$ | $79.25 \pm 0.33$ | $\mathbf{82.09 \pm 0.30}$ | $81.68 \pm 0.47$ |
| CIFAR100 (5 runs) | $57.94 \pm 0.69$ | $59.77 \pm 0.71$ | $60.95 \pm 0.64$ | $\mathbf{62.21 \pm 0.62}$ |
| Clothing1M (5 runs) | $69.60 \pm 0.85$ | $69.63 \pm 0.30$ | $71.07 \pm 0.46$ | $\mathbf{72.69 \pm 0.42}$ |
| MNLI (5 runs) | $79.19 \pm 0.53$ | $76.85 \pm 0.14$ | $\mathbf{80.89 \pm 0.31}$ | $80.28 \pm 0.33$ |
| QQP (5 runs) | $85.05 \pm 0.43$ | $\mathbf{86.30 \pm 0.41}$ | $\mathbf{86.88 \pm 0.31}$ | $\mathbf{86.99 \pm 0.49}$ |

on the Clothing1M dataset, Table 1 shows REDUCR improves the worst-class test accuracy by around $15\%$ when compared to TRAIN LOSS, the next best-performing baseline, and by around $26\%$ when compared to RHO-LOSS, the overall best-performing baseline across datasets. Figure 3a shows that REDUCR also achieves this performance in a more data efficient manner than the comparable baselines, achieving a mean worst-class test accuracy of $40\%$ within the first 10k training steps. We also observe improved efficiency on the CINIC10 dataset, as shown in Figure 3b, and the MNLI and QQP datasets as detailed in Figure 7.

The Clothing1M dataset also sees a distribution shift between the training and test dataset. In the test dataset, the worst performing class is much more prevalent than in the training dataset and as such improvements to its performance impact the average test accuracy significantly. Figure 5 shows the impact of this distribution shift as the improved performance of the model on the worst-class results in an improved average test accuracy to the state-of-the-art RHO-LOSS baseline.

## 5.2 Ablation Studies

To further motivate the selection rule in Equation (8), we conduct a series of ablation studies to show that all the terms are necessary for robust online batch selection. Figure 4a shows the performance of REDUCR on the CINIC10 dataset when the model loss, amortised class-irreducible loss and class-holdout loss terms of the algorithm were individually excluded from the selection rule. All three terms in Equation (8) are required to achieve a strong worst-class test accuracy.

Removing the Model Loss results in the worst performance in the set of ablation studies. This is because the Model Loss provides REDUCR with information about which points are currently not classified correctly by the model. By removing this term REDUCR only selects points which do well under the Class Irreducible Loss model and does not prioritise points the model has not yet learnt. Selecting points not yet learnt by the model is an important quality in online batch selection approaches and the main premise of the Train Loss baseline algorithm. Likewise by removing the Class Irreducible Loss Model term we remove the ability of the model to infer if a point can be learnt or not. In Mindermann et al. [2022], the authors note that these pretrained models enable the algorithm to pick points that are learnable and do not have label noise.

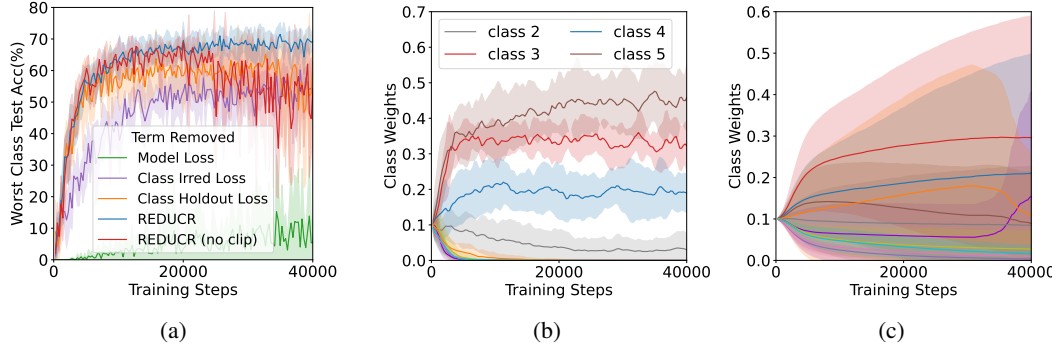

(a)          (b)          (c)

Figure 4: a) The worst-class test accuracy *decreases* when the model loss, class irreducible loss, and class-holdout loss terms are removed from REDUCR on CINIC10. Comparing REDUCR with clipping for excess losses (Algorithm 1) and REDUCR (no clip) which removes the clipping, we observe that REDUCR achieves more stable performance. We show the class weights **w** at each training step for b) REDUCR and c) REDUCR with the class-holdout loss term ablated. The ablation model fails to consistently prioritise the underperforming classes across multiple runs.

The removal of the class-holdout loss term affects the ability of REDUCR to prioritise the weights of the model correctly. In Figure 4 we compare the class weights of REDUCR and an ablation model without the class-holdout loss term. The standard model clearly prioritises classes 3, 4 and 5 during training across all 5 runs, whilst the ablation model does not consistently weight the same classes across multiple runs. We also conducted an ablation study on the clipping of the excess loss to motivate its inclusion in the algorithm, this is also shown in Figure 4a, we note that this stabilises the model performance towards the end of training and investigate further in Appendix A.7.2.

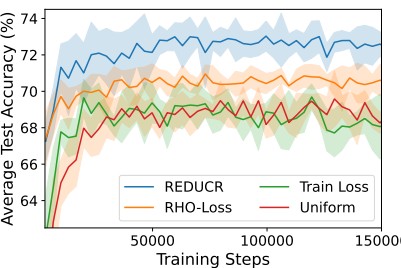

Figure 5: REDUCR improves the average test accuracy on the Clothing1M dataset.

## 5.3 Scaling up the number of classes

REDUCR can handle problems involving a large number of classes without needing to train a separate class-irreducible loss model for each class. One idea is to group the classes into superclasses, where $c \in \mathcal{G}_i, \mathcal{G}_i \in \{\mathcal{G}_i\}_{i=1}^{|G|}, \mathcal{G}_i \cap \mathcal{G}_j = \emptyset$ for $i \neq j$ and $|G| < |C|$, and solve the robust data downsampling problem over these superclasses. We test the proposed variant of REDUCR on the CIFAR100 dataset using the provided groupings with 20 superclasses in total [Krizhevsky et al., 2012]. fig. 3c shows that REDUCR outperforms the baselines in terms of the worst-**class** test accuracy, even though the robust objectives are over the superclasses. It achieves this performance in **half** the number of training steps as shown in Table 1.

## 5.4 Imbalanced Datasets

We investigate the performance of models trained using REDUCR on imbalanced datasets. We artificially imbalance the CIFAR10 training and validation datasets such that a datapoint of the imbalanced class is sampled with probability $p \in (0, 1/C]$ (referred to as the percent imbalance) and datapoints from the remaining classes are sampled with probability $(1 - p)/(C - 1)$ during model training. We conduct experiments with 0.01, 0.025 and 0.1 (which is equivalent to the balanced dataset) percent imbalance on classes 3 and 5. The results are shown in Figure 6.

We find the performance of models trained using REDUCR deteriorates less than those trained with the RHO-LOSS or UNIFORM baselines as the percent imbalance of a particular class decreases (see Figure 6). For example, when class 3 is imbalanced, in the most imbalanced case (1.0%) the median performance of REDUCR outperforms that of RHO-LOSS run by 14%. This demonstrates the effectiveness of REDUCR in prioritising the selection of data points from underrepresented classes.

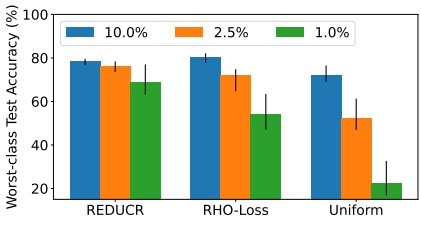 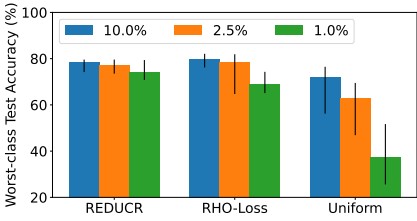

| (a) Under-sampling on class 3 | (b) Under-sampling on class 5 |

Figure 6: REDUCR significantly reduces the deterioration in the worst-class test accuracy on CIFAR10 for a) class 3 and b) class 5 when compared to the UNIFORM and RHO-LOSS baselines as the percent imbalance is decreased from 10.0% (balanced) to 1.0%. Each experiment was repeated 10 times, the median value was plotted and the error bars denote the best and worst run across 10 runs.

## 6    Conclusions, Broader Impact and Limitations

In summary, we identified the problem of class-robust data downsampling and proposed a new method, REDUCR, to solve this problem using class priority reweighting. Our experimental results indicate that REDUCR significantly enhances data efficiency during training, achieving superior test accuracy for the worst-performing class and frequently surpassing state-of-the-art methods in terms of average test accuracy. REDUCR excels in settings where the available data are class-imbalanced by prioritising the selection of points from underrepresented classes.

**Limitations.** The computational efficiency of REDUCR scales linearly with the number of classes. We propose one solution to this in Section 5.3, where we show that using groups of classes can still result in improved worst-class performance. Another solution is to use smaller model architectures for the class-irreducible loss model. [Mindermann et al., 2022] provide extensive evidence that small reference models can improve computational efficiency whilst still providing a useful signal for data selection, we leave investigation of these methods as a future research direction.

**Broader Impact.** Improving data efficiency is an important and practical problem as more machine learning models are being trained and deployed for real-world applications. Moreover it is critical to ensure the robustness of models for reliable and trustworthy machine learning. Our work proposes a new method with the goal of improving the robustness of models whilst significantly reducing the data required to achieve state of the art performance.

## Acknowledgments

This work was supported by the EPSRC New Investigator Award EP/X03917X/1; the Engineering and Physical Sciences Research Council EP/S021566/1; Google Research Scholar award and the Google Cloud Platform Credit Award. We thank Zoubin Ghahramani, Jasper Snoek, Fei Sha, Théo Galy-Fajou, Dustin Tran, Sharat Chikkerur, Xuezhi Wang and Mike Dusenberry for helpful discussions on early versions of this work.

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

# A Appendix

## A.1 Online Batch Selection Pseudo-Code

For the sake of convenience, we provide the pseudocode of the online batch selection protocol described in Section 2.

---
**Algorithm 2** Online batch selection

---
1: **Input:** data pool $\mathcal{D}$, number of training steps $T$, stochastic gradient descent algorithm SGD, a loss function $\mathcal{L}$
2: **for** $t = 1$ to $T$ **do**
3:      Sample batch $B_t$ randomly from $\mathcal{D}$
4:      $b_t = \text{SelectBatch}(B_t, \theta_t)$
5:      $\text{L} = \sum_{(x_i, y_i) \in b_t} \mathcal{L}[y_i | x_i, \theta_t]$
6:      $\theta_{t+1} = \text{SGD}(L, \theta_t)$
7: **end for**

---

## A.2 Class Irreducible Loss Model Training Pseudo-Code

Here we detail the pseudo-code for training the class irreducible loss model described in Section 4.3

---
**Algorithm 3** Class Reference Model Training

---
1: **Input:** holdout dataset $\mathcal{D}_{ho}$, number of training steps $T$, stochastic gradient descent algorithm SGD, a loss function $\mathcal{L}$, a specific class $c$
2: **for** $t = 1$ to $T$ **do**
3:      $B_{ho} \sim \text{Uniform}(\mathcal{D}_{ho})$
4:      $L = \sum_{(x_i, y_i) \in B_{ho}} (1 + \gamma \mathbb{I}[c = y]) \mathcal{L}[y_i | x_i, \phi_t]$
5:      $\phi_{t+1} = \text{SGD}(L, \phi_t)$
6: **end for**
7: **Return** Class-irreducible loss model parameters $\phi_T$

---

## A.3 The Amortised Class Irreducible Loss Model Approximation

The amortised class irreducible loss model is an important component in REDUCR as shown by our ablation study in Figure 4a. For each class $c \in [C]$, we approximate the second term of Equation (6), $\log p(y|x, \mathcal{D}_t, \mathcal{D}_{ho}^{(c)})$ via the model trained using Algorithm 3. This approximation has two steps: firstly we remove dependence of the class irreducible model loss on the training dataset at time $t$. A similar approximation is heavily explored by Mindermann et al. [2022] in Section 3 and Section 4 of their paper; in Appendix D of their work they show that this approximation is important for the stable training of RHO-LOSS. The approximation also aligns RHO-LOSS and REDUCR with other methods in the literature such as Xie et al. [2023], Oren et al. [2019] which similarly use a reference model that does not vary during training.

Secondly we up-weight data points in the loss function when their label $y \in [C]$ matches that of the specific class $c$. Unlike RHO-LOSS we cannot approximate the class irreducible loss as $\log p(y|x, \mathcal{D}_t, \mathcal{D}_{ho}^{(c)}) \approx \log p(y|x, \mathcal{D}_{ho}^{(c)})$ as this is a trivial model only trained on points with labels from a single class and thus does not provide a suitable signal to guide point selection. We interpret the original class irreducible loss $\log p(y|x, \mathcal{D}_t, \mathcal{D}_{ho}^{(c)})$, as an expert model for class $c$ as this model trains on extra points only sampled from that class, $\mathcal{D}_{ho}^{(c)}$. In our approximation we train on the holdout dataset which does not have excess examples of points from class $c$. We justify our up-weighting of points as a form of importance weighting [Shimodaira, 2000], where by up-weighting points with labels in a specific class we are calculating a Monte Carlo approximation of the loss under a distribution in which points from class $c$ are more prevalent.

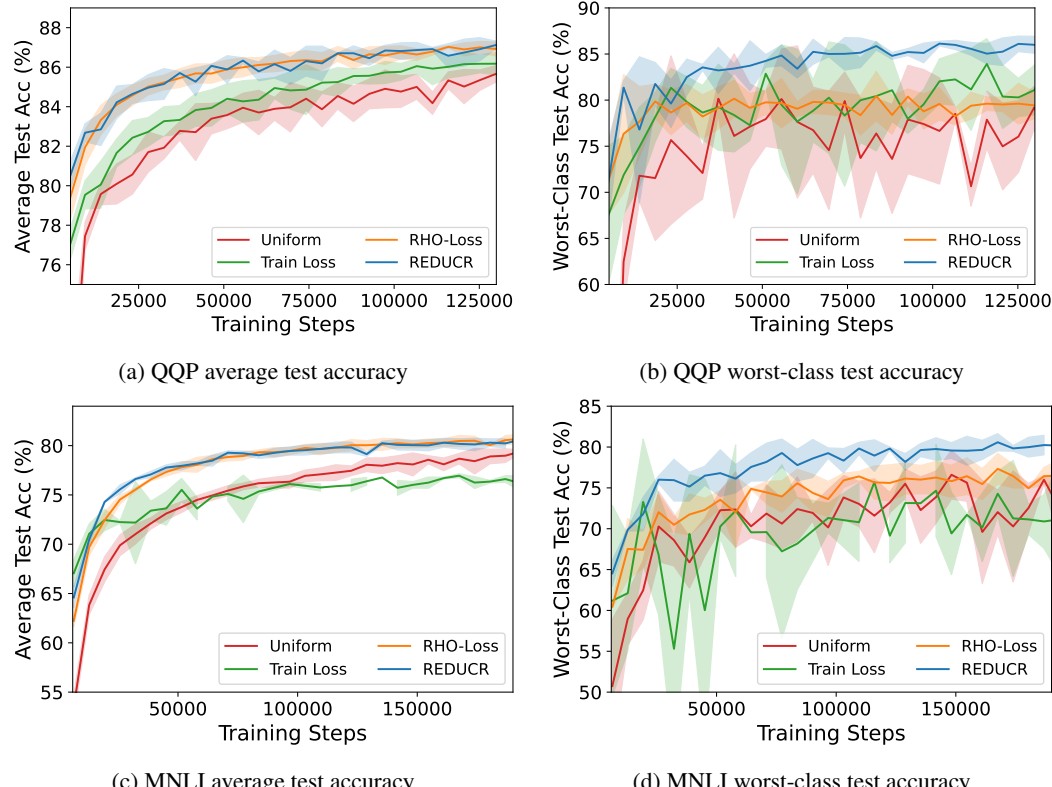

(a) QQP average test accuracy         (b) QQP worst-class test accuracy

(c) MNLI average test accuracy         (d) MNLI worst-class test accuracy

Figure 7: REDUCR improves the worst-class test accuracy on the MNLI and QQP text datasets whilst maintaining strong average test accuracy performance when compared with the TRAIN LOSS, RHO-LOSS and UNIFORM baselines. On both datasets REDUCR matches the next best performing baseline's mean result across runs approximately 100k training steps earlier.

### A.4 The Effect of the Class-Holdout Loss on the Selection of Points

The class-holdout loss term only affects the selection of points at each iteration $t$ through the selection of the weights $\mathbf{w}_t$. As it does not depend upon the candidate point $(x, y) \in B_t$ and the weights sum to one we can remove it from line 6 of Algorithm 1 and only include it in line 7 when we update the class weights. Similarly as the model loss does not depend upon the class $c$ we can write the selection score as

$$\sum_{c \in C} w_{t,c} \log p(\mathbf{y}_{ho}^{(c)} | \mathbf{x}_{ho}^{(c)}, \mathcal{D}_t \cup (\{x, y\})) = \mathcal{L}[y|x, \theta_t] - \sum_{c \in C} w_{t,c}(\mathcal{L}[y|x, \theta_t^{(c)}]) - \sum_{c \in C} w_{t,c}(\mathcal{L}[\mathbf{y}_{ho}^{(c)} | \mathbf{x}_{ho}^{(c)}, \theta_t]).$$

(10)

### A.5 Experiment Details

We provide the full code base anonymised for review purposes as part of the supplementary material.

**CIFAR10** used half the training dataset (25k points) as a holdout validation dataset for training the amortised class-irreducible loss models and calculating the class-holdout loss during the robust online batch selection. We used the remaining 25k points as a training dataset and the provided test dataset (10k) for testing.

**CINIC10** used the provided validation dataset for both the class holdout loss and amortised class irreducible loss models.

**CIFAR100** used the provided validation dataset for both the class holdout loss and amortised class irreducible loss models.

**Clothing1M.** The dataset consists of 1 million images labelled automatically using the keywords in its surrounding text. The dataset consists of 72k 'clean' images whose labels have been hand checked, 50k, 13k and 9k are respectively sorted into a clean training, validation, and test sub-dataset. To train

the amortised class irreducible loss models we use 100k points randomly sampled from the union of the validation, clean and noisy training datasets. We calculate the class-holdout loss term and validation performance during training using the clean validation dataset. Figure 8 shows how the class distribution changes between train and test times.

**MNLI.** The dataset [Williams et al., 2018] consists of 412k labeled sentence pairs; similarly to Sagawa et al. [2020] we split these sentence pairs into a train (206k), validation (164k), and labelled test (41k) dataset.

**QQP.** The dataset consists of 431k labeled sentence pairs; we remove points from class 1 to further imbalance the dataset resulting in 22% of the dataset labelled class 1. We split the remaining points into a train (148k), validation (67k), and labelled test (40k) dataset. We do not adjust the balance of the test dataset.

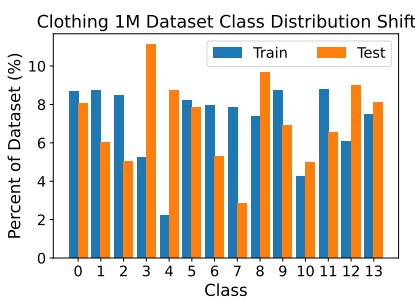

Figure 8: Clothing1M train-test class distribution shift. The number of points in classes 4 and 7 change dramatically between the train and test sets.

**ResNet-18** used for the Clothing1M experiments is the pretrained model available via the Torchvision [Marcel and Rodriguez, 2010] model library. For the CIFAR10 and CINIC10 experiments we use the adapted ResNet-18 architecture detailed in Mindermann et al. [2022] Appendix B.

**Train Loss** baseline is taken from Loshchilov and Hutter [2016] where points from the large batch $B_t$ are sampled with probability

$$p_i \propto \frac{1}{\exp(\log(s)/|B_t|)^i}.\tag{11}$$

Here $p_i$ is the point with the $i^{th}$ highest training loss in the large batch. We set the selection pressure parameter $s_e = 100$ and do not vary this during training as per the Experiments in Section 6. of Loshchilov and Hutter [2016].

**Compute Resources and Data Sources.** All models were trained on GCP NVIDIA Tesla T4 GPUs. The Image datasets were sourced from pytorch via the torchvision datasets package `https://pytorch.org/vision/stable/datasets.html`, the NLP datasets were sourced from huggingface, `https://huggingface.co/datasets/nyu-mll/glue`.

**Data Augmentation** was applied to the training dataset during online batch selection and validation dataset during the training of the amortised class-irreducible loss model. We apply a random crop and random flip to the images.

### A.6 Results with worst-class checkpointing

In Table 3 and Table 4 we show the worst-class and average test accuracy respectively, when the UNIFORM, TRAIN LOSS and RHO-LOSS baselines use worst-class validation accuracy to checkpoint the model during training. REDUCR still outperforms or matches the best baseline performance across all datasets. In the cases where REDUCR matches the performance of the best performing baseline, it does so in a more data efficient manner. Figure 7b and Figure 7d show the mean and standard deviation worst-class test accuracy across multiple runs on the QQP and MNLI datasets. REDUCR matches the best mean performance of the best performing baseline almost 100k training steps earlier on both datasets.

### A.7 Additional Experimental Results

In Appendix A.7.1 we show the per class weights for the Clothing1M dataset, whilst in Appendix A.7.2 we analyse the effect of the clipping term and provide some intuition behind its inclusion in the algorithm.

### A.7.1 Clothing1M Training Weights

The Clothing1M dataset is imbalanced with respect to class 4. Figure 9 shows that REDUCR is able to consistently identify and weight the underrepresented class across model runs.

| Dataset | Worst-Class Test Accuracy (%) $\pm 1$ std | | | |
|---|---|---|---|---|
| | UNIFORM | TRAIN LOSS | RHO-LOSS | REDUCR |
| CIFAR10 (10 runs) | $75.01 \pm 1.37$ | $79.32 \pm 1.35$ | $81.23 \pm 1.18$ | $\mathbf{83.29 \pm 0.84}$ |
| CINIC10 (10 runs) | $70.86 \pm 1.23$ | $68.89 \pm 0.86$ | $73.44 \pm 1.16$ | $\mathbf{75.30 \pm 0.85}$ |
| Clothing1M (5 runs) | $39.23 \pm 5.41$ | $49.02 \pm 2.32$ | $32.19 \pm 9.83$ | $\mathbf{53.91 \pm 2.42}$ |
| MNLI (5 runs) | $76.88 \pm 1.21$ | $75.75 \pm 0.56$ | $\mathbf{78.04 \pm 1.73}$ | $\mathbf{79.45 \pm 0.39}$ |
| QQP (5 runs) | $84.50 \pm 0.56$ | $\mathbf{85.49 \pm 1.32}$ | $82.60 \pm 1.12$ | $\mathbf{86.61 \pm 0.49}$ |

Table 3: Worst-class test accuracy, when the RHO-LOSS and TRAIN LOSS baselines are checkpointed using their worst-class validation error during training.

| Dataset | Average Test Accuracy (%) $\pm 1$ std | | | |
|---|---|---|---|---|
| | UNIFORM | TRAIN LOSS | RHO-LOSS | REDUCR |
| CIFAR10 (10 runs) | $85.09 \pm 0.52$ | $87.74 \pm 0.50$ | $\mathbf{89.43 \pm 0.57}$ | $\mathbf{90.02 \pm 0.44}$ |
| CINIC10 (10 runs) | $79.57 \pm 0.75$ | $78.21 \pm 0.57$ | $81.28 \pm 0.54$ | $\mathbf{81.68 \pm 0.47}$ |
| Clothing1M (5 runs) | $69.60 \pm 0.85$ | $69.46 \pm 0.43$ | $70.63 \pm 0.87$ | $\mathbf{72.69 \pm 0.42}$ |
| MNLI (5 runs) | $78.85 \pm 0.38$ | $78.50 \pm 0.33$ | $\mathbf{80.50 \pm 0.45}$ | $\mathbf{80.28 \pm 0.33}$ |
| QQP (5 runs) | $85.23 \pm 0.36$ | $\mathbf{86.24 \pm 0.26}$ | $\mathbf{86.75 \pm 0.37}$ | $\mathbf{86.99 \pm 0.49}$ |

Table 4: Average test accuracy, when the RHO-LOSS and TRAIN LOSS baselines are checkpointed using their worst-class validation error during training.

### A.7.2 Clipped Excess Loss Ablation Experiments

To further understand the effects of clipping in the algorithm we analyse the selection score of the selected points with and without clipping. As detailed in Appendix A.4 the class-holdout loss only affects the selection of points via the weights $\mathbf{w}_t$ at each time step, as such we record only the excess loss (the difference between the model loss and class irreducible loss). Figure 10 shows the quantiles of the weighted sum of the excess losses of points selected at each training step for the non-clipped and clipped model respectively. When the excess loss is clipped, Figure 10a shows the selection scores smoothly decrease throughout training as the model loss improves. Without clipping the excess loss decreases smoothly at the beginning of training and then shows unstable behaviour across runs later in training.

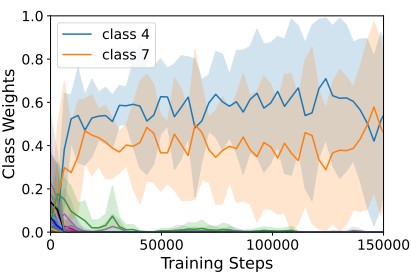

Figure 9: Clothing1M class weights

In practice we select multiple points per batch by selecting the points with the top k selection scores. When multiple points have the same score, points are selected at random. We note that the clipping does not reduce the excess loss of the selected points to zero where points would be selected randomly to make up the batch.

Intuitively we posit that the clipping reduces the effect of clashing amortised class irreducible loss models in the weighted sum across the $|C|$ selection rules. The amortised class irreducible loss models are trained such that they are an expert in a specific class $c$. In some cases a model being an expert in a specific class $c'$ may result in it being a poor predictor of classes $C \setminus c'$. Even if this expert has a small weight $w_{t,c'}$ large losses may still propagate into the selection of points. Clipping the excess loss prevents a point from being down-weighted in the weighted sum of class specific selection scores by a specific class too much.

### A.7.3 Alternative Solutions to Robust Data Downsampling

In Section 4.1 we introduce an online solution to the robust data downsampling problem based upon the multiplicative weights method. In this section we empirically evaluate a variation of REDUCR where we solve the maximin optimisation problem directly. At each time step $t$ we select a small

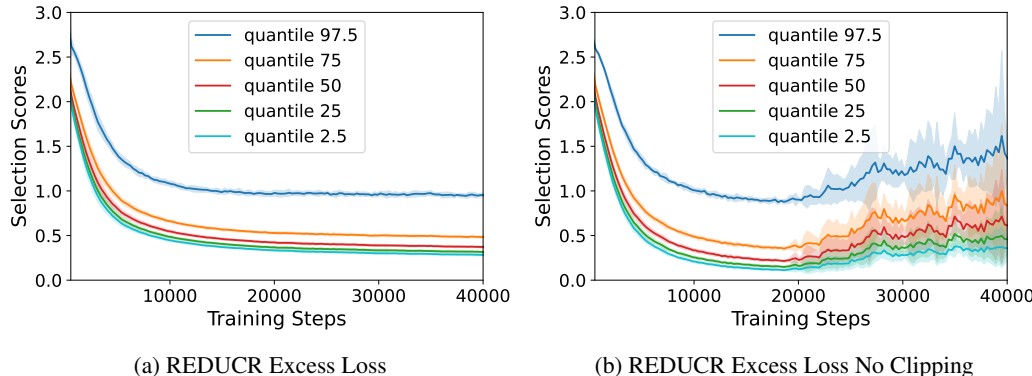

(a) REDUCR Excess Loss

(b) REDUCR Excess Loss No Clipping

Figure 10: The quantiles of the excess loss of points selected at each training step with (a) and without clipping (b) of the excess loss term

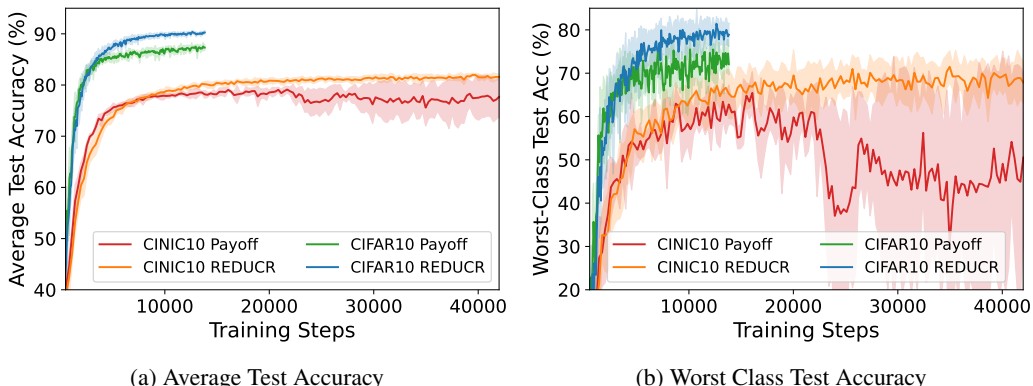

(a) Average Test Accuracy

(b) Worst Class Test Accuracy

Figure 11: We compare REDUCR with Algorithm 4 in which the maximin optimisation problem approximated by solving the payoff matrix directly at each step $t$, this is labelled Payoff. REDUCR consistently outperforms this approach both in terms of (a) average test accuracy and (b) worst class test accuracy.

batch $b_t \subset B_t$

$$b_t = \{(x,y)\} = \operatorname*{argmax}_{(x,y) \in B_t} \min_{c \in [C]} \mathcal{L}[y|x, \theta_t] - \mathcal{L}[y|x, \theta_t^{(c)}] - \mathcal{L}[\mathbf{y}_{ho}^{(c)}|\mathbf{x}_{ho}^{(c)}, \theta_t], \tag{12}$$

for clarity we once again write the selection rule in terms of a single point. As $|B_t| << |\mathcal{D}|$ and both $[C]$ and $B_t$ are discrete sets it is feasible to solve the optimisation problem directly. To do this we first minimize the selection score for each datapoint $(x, y)$ with respect to a class $c$ and then select the $k$ datapoints with the greatest minimum score. The full algorithm is shown in Algorithm 4. We compare this approach against REDUCR on the CIFAR10 and CINIC10 datasets. The results can be seen in Figure 11 where REDUCR outperforms the direct optimisation approach in terms of both average test accuracy and worst class test accuracy.

---

**Algorithm 4** Robust Data Downsampling Approximated by Directly Solving the Payoff Matrix

---

1: **Input:** data pool $\mathcal{D}$, holdout data $\mathcal{D}_{ho} = \bigcup_{c \in C} \mathcal{D}_{ho}^{(c)}$, total timesteps $T/k$, small batch size $k$
2: **for** $t \in [T/k]$ **do**
3:     Sample batch $B_t$ randomly from $\mathcal{D}$
4:     $b_t = \operatorname*{argmax}_{b \in B_t : |b| = k} \sum_{(x,y) \in b} \min_{c \in |C|} \mathcal{L}[y|x, \theta_t] - \mathcal{L}[y|x, \theta_t^{(c)}] - \mathcal{L}[\mathbf{y}_{ho}^{(c)}|\mathbf{x}_{ho}^{(c)}, \theta_t]$    ▷ Select points
    with top k min scores
5:     $\theta_{t+1} = \text{SGD}(\theta_t, b_t)$
6: **end for**

---

## A.8 Experiments on ConvNext Architecture

We repeated the Clothing1M experiments using the `facebook/convnext-tiny-224` ConvNext model Liu et al. [2022] from HuggingFace, the results are shown in Figure 12. Here we note REDUCR maintains strong performance in terms of the average test accuracy. The mean worst-class test accuracy outperforms the other baselines, however is not statistically significant.

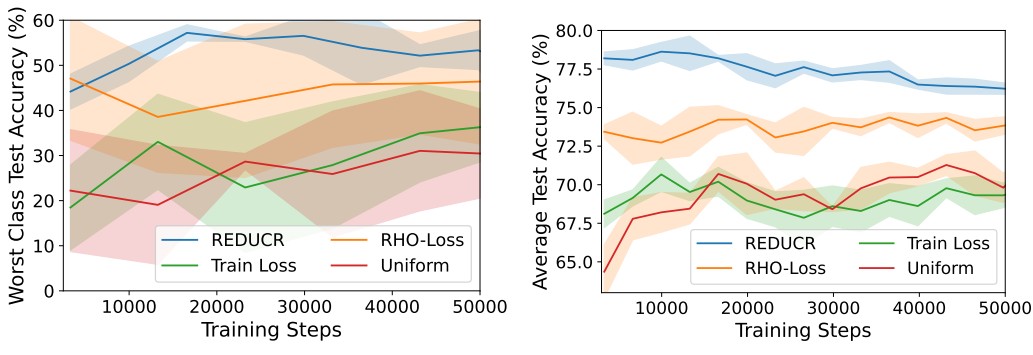

(a) Worst-Class Test Accuracy (3 seeds)  (b) Average Test Accuracy (3 seeds)

Figure 12: a) REDUCR outperforms the relevant baselines in terms of the mean worst-class accuracy using the *facebook/convnext-tiny-224* ConvNext Model [Liu et al., 2022], we ran each experiment for 3 seeds with no hyperparameter tuning. b) REDUCR continues to outperform all the baseline algorithms in terms of the average test accuracy on the Clothing1M dataset when training the ConvNext model. Whilst the worst-class test accuracy is noisy, the improvement REDUCR makes across multiple poorly performing classes results in a large performance difference between it and the next best performing baseline.

## A.9 Amortised Class Irreducible Loss Models

In Figure 13 we compare the average expert class ($c \in C$) test accuracy and non-expert class ($c' \in C \setminus \{c\}$) test accuracy across different values of $\gamma$ for the amortised class-irreducible loss model train on CIFAR10. For the model to be an expert in one class it loses performance in the non-relevant classes. To avoid the problems described in Appendix A.7.2 we selected $\gamma = 9$ for the image datasets as the performance of the non-expert class did not suffer too much.

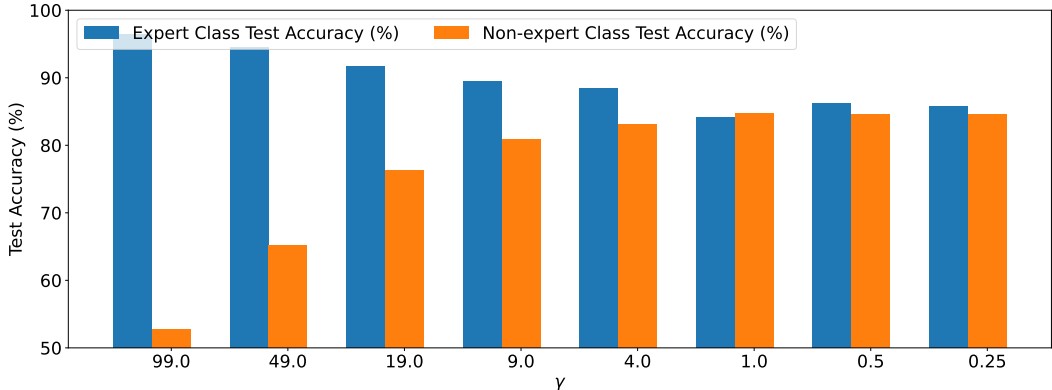

Figure 13: Class-irreducible loss model test accuracies on the expert class and non-expert classes. Class-irreducible loss models are trained using gradient weights $\gamma \in \{0.25, 0.5, 1.0, 4.0, 9.0, 19.0, 49.0, 99.0\}$.

## A.10 Hyperparameter Tuning

In this section, we test the sensitivity of REDUCR with respect to the hyperparameters introduced. In particular, we investigate the sensitivity of the learning rate $\eta$, used for target model training; the gradient weight $\gamma$, used for class-irreducible loss model training; the fraction of datapoints selected for target model training $|b_t|/|B_t|$, for a constant selected batch size $|b_t|$; and the frequency with

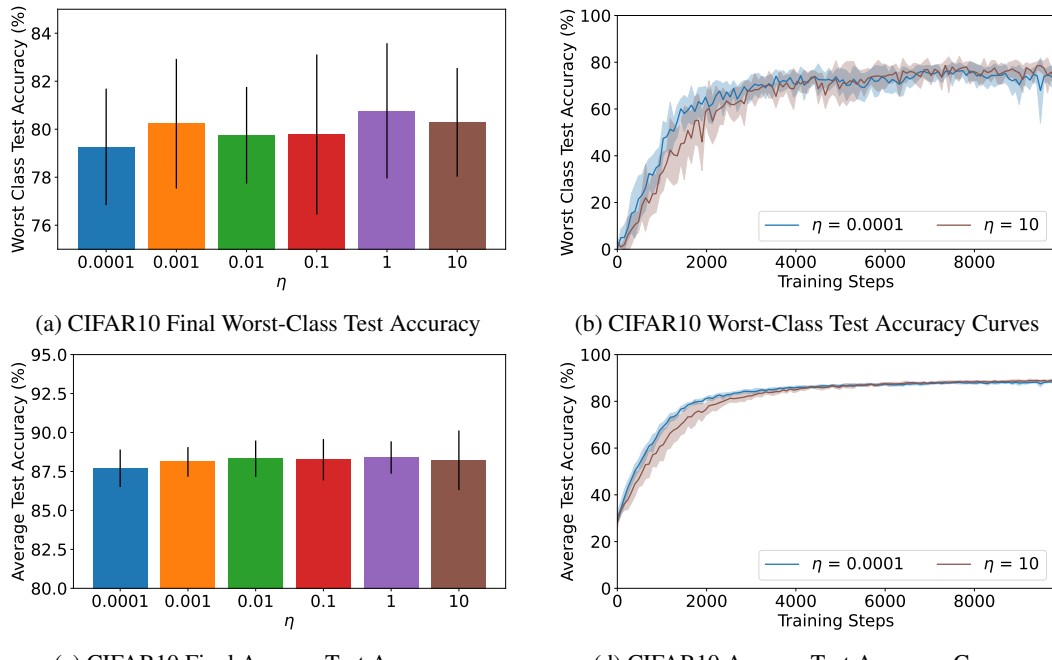

(a) CIFAR10 Final Worst-Class Test Accuracy

(b) CIFAR10 Worst-Class Test Accuracy Curves

(c) CIFAR10 Final Average Test Accuracy

(d) CIFAR10 Average Test Accuracy Curves

Figure 14: The final average and worst-class test accuracy are not sensitive to the value of $\eta$ on the CIFAR10 dataset. When using smaller values of $\eta$ both the average and worst class test accuracy attain higher performance at an earlier training step.

which the class holdout loss term is updated during training. All experiments in this section use the CIFAR10 dataset. We use ResNet-18 target models, trained using $\eta = 10^{-4}$ with a fraction of datapoints selected of $|b_t|/|B_t| = 0.10$, and ResNet-18 class irreducible loss models trained using $\gamma = 9$ unless otherwise stated.

We find that REDUCR is not sensitive to the learning rate $\eta$ or the frequency of the class holdout loss term updates. We find that the performance of REDUCR is sensitive to the gradient weight $\gamma$ at high values. Finally, we find that REDUCR is not sensitive to the fraction of data points selected for target model training (referred to as the percent train) for intermediate values of percent train, though performance is poor for very low fractions and very high fractions recovers uniform selection as $b_t = B_t$ when $|b_t|/|B_t| = 1.0$ and $B_t$ is sampled uniformly from the dataset.

In summary, REDUCR is largely insensitive to the values of the newly introduced hyperparameters when trained on the CIFAR10 dataset. Sensitivity analyses on additional datasets are needed to increase the robustness of these findings. However, a gradient weight of $\gamma = 9$ and a percent train of 0.10 perform well without additional hyperparameter tuning for several datasets, as shown in Table 1 and Table 2 which tentatively supports the robustness of these findings.

### A.10.1 Learning Rate $\eta$

First, we perform a sensitivity analysis on the learning rate $\eta$ for values $\eta \in \{10^{-4}, 10^{-3}, 10^{-2}, 10^{-1}, 10^{0}, 10^{1}\}$. The experimental results, shown in Figure 14, demonstrates that smaller values of $\eta$ result in a faster improvement of the average and worst class test accuracy during training, although the final model performance is similar for all values of $\eta$ investigated. In practice, appropriately small values of $\eta$ should be used in order to reduce computational cost. Note that what constitutes an appropriately small value of $\eta$ depends on the scale of losses in a particular domain. Initial target model training runs can be done to identify a value of $\eta$ for which class weights do not prematurely concentrate on one class $\eta$.

### A.10.2 Gradient Weight $\gamma$

To investigate the sensitivity of the gradient weight $\gamma$ on the performance of REDUCR. We train sets of class-irreducible loss models for each $\gamma \in \{0.25, 0.5, 1.0, 4.0, 9.0, 19.0, 49.0, 99.0\}$ and train a model for each set of class-irreducible loss models. The results, shown in Figure 15, show that

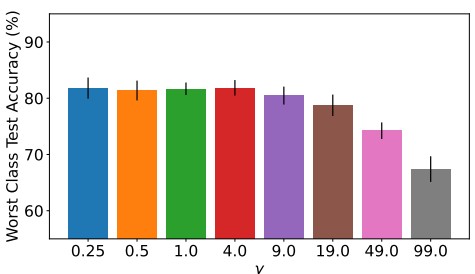
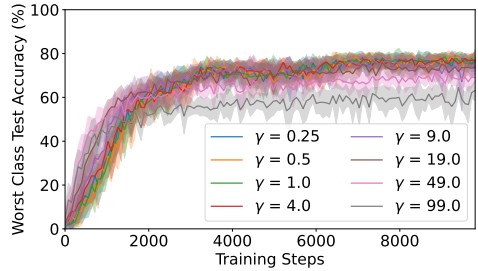

(a) CIFAR10 Final Worst-Class Test Accuracy

(b) CIFAR10 Worst-Class Test Accuracy Curves

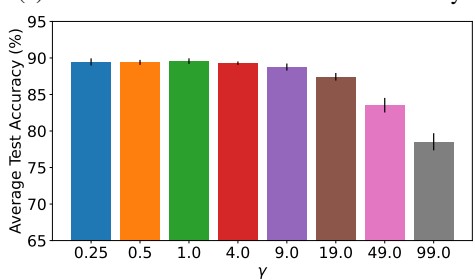
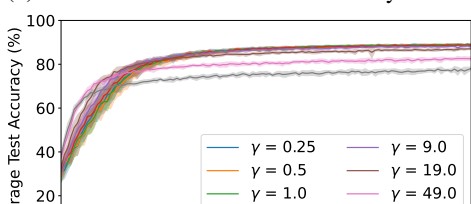

(c) CIFAR10 Final Average Test Accuracy

(d) CIFAR10 Average Test Accuracy Curves

Figure 15: **Average and worst-class test accuracy are sensitive to the value of $\gamma$ on the CIFAR10 dataset**, though this likely reflects longer convergence times for class-irreducible loss model training when using larger values of $\gamma$.

values of the gradient weight above $9.0$ result in faster improvement of the average and worst class test accuracy early in training, although these models converges to a lower average and worst class test accuracy at the end of training. Higher gradient weights also increase the variance of the class-irreducible training loss, models trained with higher weights thus required a greater number of gradient descent steps to converge to a suitable class-irreducible loss model. We found class irreducible loss models trained with gradient weights $\gamma \in \{19.0, 49.0, 99.0\}$ are selected via checkpointing before the model has converged and suspect this is a contributing factor in the poor test accuracy we observed.

This finding highlights the trade-off between fast target model training, which requires a large gradient weight; fast class irreducible loss model training, which requires a smaller gradient weight; and strong final performance of the model. Final average and worst class test accuracy is similar for all models trained with gradient weights $\gamma \in \{0.25, 0.5, 1.0, 4.0, 9.0\}$.

### A.10.3 Fraction of Selected Datapoints

We perform a sensitivity analysis on the fraction of datapoints selected for target model training $|b_t|/|B_t|$, referred to as the percent train hyperparameter. We use a constant small selected batch size $|b_t|$ and vary the large batch size $|B_t|$ in order to vary the fraction of datapoints selected for target model training. In this setting, a smaller percent train allows REDUCR to select from a greater number of candidate datapoints at each training step, which results in the selection of datapoints with larger weighted reducible loss. Since datapoints with larger weighted reducible loss are those from which a model can learn the most [Mindermann et al., 2022], we expect a smaller percent train to result in a faster improvement in target model performance.

We train target models using REDUCR for each percent train and large batch size pair $(|b_t|/|B_t|, |B_t|)$ in $\{(0.05, 640), (0.10, 320), (0.15, 216), (0.20, 160), (0.25, 128)\}$ and present the results in Figure 16. We find that a percent train of $0.05$ attains lower final worst-class and average test accuracy, despite having most candidate datapoints to select from. This is surprising and is in contradiction with the intuition provided above. Furthermore, percent trains $\{0.1, 0.15, 0.2, 0.25\}$ attain similar average test accuracy at the end of training, though larger percent trains attain slightly higher worst-class test accuracy at the end of training.

These results demonstrate that the performance of REDUCR is largely insensitive to the percent train hyperparameter for a constant selected batch size. In practice, a selected batch size $|b_t|$ should first be chosen such that loss gradient estimates have a low variance, and then a large batch size $|B_t|$ should

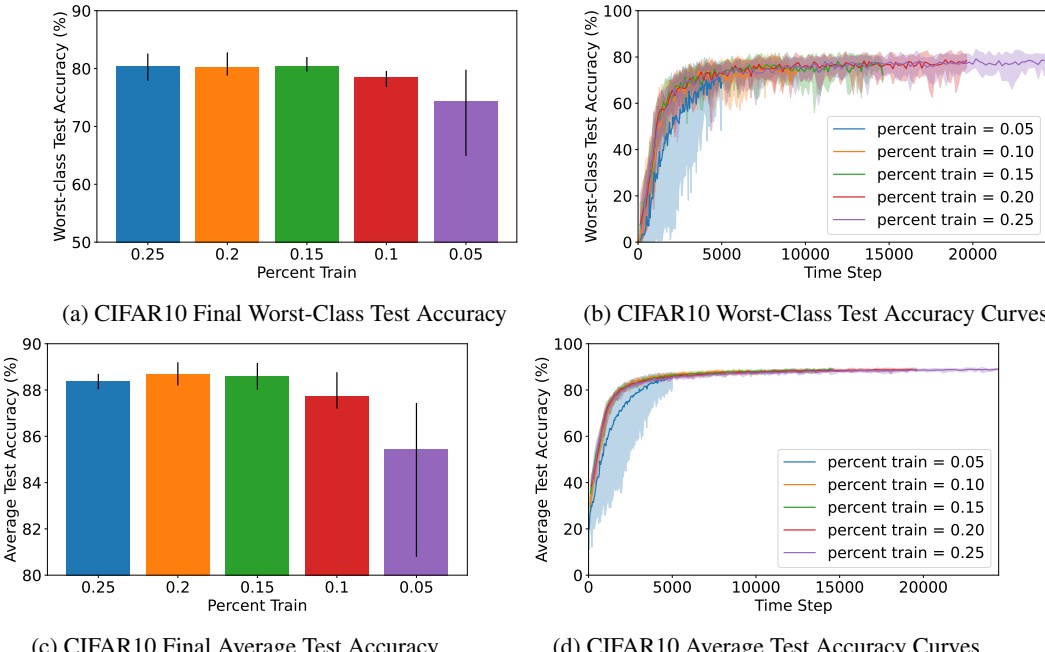

(a) CIFAR10 Final Worst-Class Test Accuracy

(b) CIFAR10 Worst-Class Test Accuracy Curves

(c) CIFAR10 Final Average Test Accuracy

(d) CIFAR10 Average Test Accuracy Curves

Figure 16: REDUCR is broadly insensitive to changes of the percent train hyperparameter, $|b_t|/|B_t|$ on the CIFAR10 dataset. The final worst class and average test accuracy is statistically similar across a variety of values of the percent train hyperparameter. For a small value of percent train 0.05, the final worst class and average test accuracy suffers and REDUCR requires more gradient descent steps to match the performance of models trained with larger percent train values. The plots show the mean and standard deviation across 10 runs.

be chosen such that the percent train is an intermediate value for example 0.10. These results also suggest that selecting datapoints with the very largest weighted reducible loss for model training may not be most appropriate for improving model performance. Instead of top-$k$ selection a more nuanced approach that accounts for the joint distribution over all points in the selected batch could be used.

### A.10.4 Frequency of Class Hold-out Loss Updating

The class holdout loss term is updated at the beginning of each training epoch using the full holdout dataset. As each epoch consists of multiple gradient descent steps the actual performance of the target model on the holdout dataset will vary before the class holdout loss term is recalculated. In this section we investigate an alternative method in which the class holdout loss term is updated at every gradient descent step.

It is computationally expensive to update the class holdout loss term using the full holdout dataset at every gradient descent step. Therefore, we propose an alternative fast updating method, which only uses a batch of holdout points to estimate the term at each timestep. The class holdout loss computed in this manner is noisy. Therefore, we use an exponentially-weighted moving average of the class holdout losses from previous timesteps to produce a smoother signal. Specifically, a batch of size 320 is sampled uniformly at random from the holdout dataset at each training step. Losses are then computed for each datapoint in the sampled batch using the current target model. Finally, for each class $c \in [C]$, losses of datapoints of class $c$ in the sampled batch are averaged and used to update a debiased exponentially-weighted moving average with decay parameter $a \in [0, 1]$.

We perform experiments using exponentially-weighted moving averages with decay parameters $a \in 0.9, 0.99$ for fast updating of the class holdout loss term. The results shown in Figure 17 demonstrate REDUCR is not improved in a statistically significant manner by the more complex exponentially-weighted moving average approach, both in terms of the final average and worst class test ac-curacies or the number of training timesteps required to reach these accuracies.

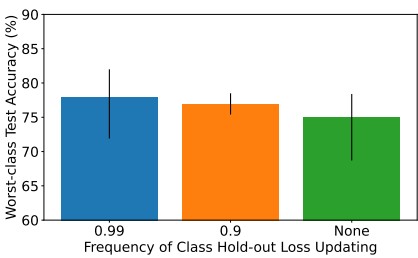

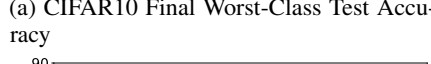

(a) CIFAR10 Final Worst-Class Test Accuracy

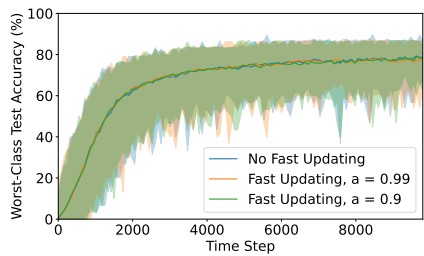

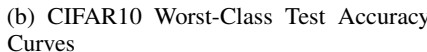

(b) CIFAR10 Worst-Class Test Accuracy Curves

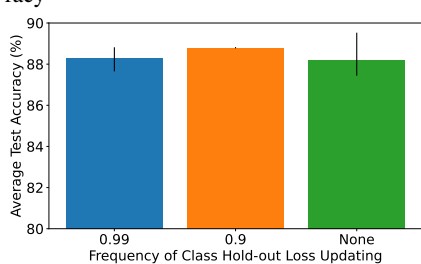

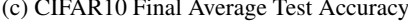

(c) CIFAR10 Final Average Test Accuracy

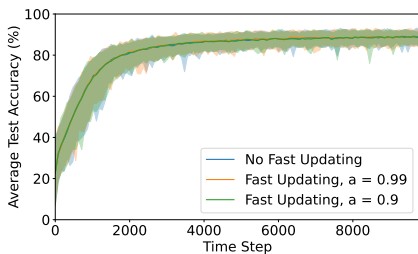

(d) CIFAR10 Average Test Accuracy Curves

Figure 17: Test accuracies are not sensitive to the frequency with which class hold-out losses are updated. Plots show minimums, medians and maximum across 10 seeds.

## A.11 Highly Imbalanced Datasets

We also conduct experiments with 0.25% and 0.5% percent imbalances on classes 3 and 5. However, (class) irreducible loss models and target models only receive 6.25 and 12.5 datapoints of the imbalanced class during one training epoch (in expectation) with percent imbalances of 0.25% and 0.5% respectively. As a result, too few datapoints of the imbalanced class are seen during model training to achieve good performance on the imbalanced class.

