# OpenReview forum: "REDUCR: Robust Data Downsampling using Class Priority Reweighting"
_NeurIPS.cc/2024/Conference — NeurIPS 2024 poster_

### Official Review · Reviewer_75mt · 2024-07-08

**Soundness:** 3
**Presentation:** 3
**Contribution:** 2
**Rating:** 6
**Confidence:** 3

**Summary:**

This paper proposes a data downsampling method called REDUCR which is robust to class imbalance and distribution shits. In particular, REDUCR reduces the scale of the training data while emphasizing the datapoints with the worst generalization performance. Experiments on 6  benchmarks demonstrate the effectiveness of the method.

**Strengths:**

* This paper is well-written and well-organized.
* The motivation of this paper is clear.

**Weaknesses:**

* In the abstract, the authors claim that REDUCR is robust to distribution shifts. However, the experiment part does not include relevant datasets with different types of distribution shifts between the training and the test samples.
* Although the technical novelty might be somewhat limited, I like the fact that the method is rather straightforward and effective.

**Questions:**

* Could you provide the numerical results on datasets with distribution shifts?
* Could you discuss other kinds of model architectures such as ConvNeXt and Swin to verify the generalizability of REDUCR?

**Limitations:**

The authors claim the limitation of this work and that there are no negative societal impacts to be highlighted.

---

> ### Author Rebuttal · Authors · 2024-08-07
>
> We thank the reviewer for their review, recognising the strong motivation behind our work, and that our paper is well-written. In response to the weaknesses and questions:
>
> > W1/Q1: REDUCRs Performance under Distribution Shifts
>
> In response to the reviewer’s first question, we direct the reviewer to line 272 of the Key Results Section. Here we note that the Clothing 1M dataset sees a distribution shift between the training and test set where the worst performing class is more prevalent in the test dataset. We will emphasise this in the text and include a graph to show how the number of points per class changes between the training and test set of the Clothing1M dataset in the appendix of the paper (see the rebuttal pdf).
>
> We also record the worst-class performance on the test set of all our datasets. The worst possible class distribution shift (defined in terms of groups in [1]) between the training and test set is one where the entire test set consists of only points from the worst performing class. As such one can consider the worst-class test accuracy as a lower bound on the performance under any class distribution shift. In Table 1, we show that REDUCR outperforms all the baselines on this metric. We will adjust the text to make this clearer in the manuscript by adding at the end of the Experiment setup (before Section 5.1):
>
> *“Finally, it is important to note that we analyse the worst-class test accuracy which can be interpreted as a lower bound on a model’s performance under all class distribution shifts. This is because the worst possible distribution shift between the training and test set is one where the entire test set consists of only points from the worst performing class.”*
>
> Whilst other types of distribution shift exist, class distribution shift is an important subset and can drastically affect the performance of algorithms - the worst-class test accuracy varies dramatically from the average test accuracy on certain datasets. Robust online batch selection to address a wider variety of types of distribution shift remains an open problem and is an interesting direction for consideration in future works.
>
> > W2: Novelty
>
> To the best of our knowledge our work is the first to propose and address the question of robust online batch selection. Whilst the use of reference models to help point selection and a weighting strategy to improve the worst class performance are both present in the literature, using those ideas to solve the problem is non-trivial and is a novel contribution of our paper. This can be seen through the class holdout loss term and class irreducible loss model which arise only in the robust online batch selection setting and are unique to our work.
>
> > Q2: Generalizability of REDUCR
>
> We thank the reviewer for their recommendation and have included experiment results using the Clothing1M dataset and the ConvNext architecture in the attached pdf.
>
> We repeated the Clothing1M results using the *facebook/convnext-tiny-224*, ConvNext model architecture [2] from HuggingFace. The results are presented in the attached pdf. We note that REDUCR continues to match or outperform all the baseline approaches on the dataset. We ran each experiment for 3 seeds and used the hyperparameters from our original Clothing1M experiment. These are promising initial results and further hyperparameter tuning specific to the model architecture will improve the performance further.
>
> We have addressed all of the reviewers points, run additional experiments to further verify the generalizability of REDUCR and included several clarifying remarks in the manuscript. In light of this we ask that the reviewer reconsider and increase their score.
>
> ***References***
>
> [1] Shiori Sagawa et al. Distributionally robust neural networks for group shifts: On the importance of regularization for worst-case generalization. Internation Conference on Learning Representations, 2020.
>
> [2] Zhuang Liu, Hanzi Mao, Chao-Yuan Wu, Christoph Feichtenhofer, Trevor Darrell, and Saining Xie. A convnet for the 2020s. In Proceedings of the IEEE/CVF conference on computer vision and pattern recognition, pages 11976–11986, 2022.

---

> > ### Comment · Reviewer_75mt · 2024-08-11
> >
> > Thank you for your response. It has addressed most of my concerns. Therefore, I chose to raise my score.

---

> > > ### Author Response · Authors · 2024-08-12
> > >
> > > Thank you for your feedback on our work and for raising your score.

---

### Official Review · Reviewer_9L5X · 2024-07-12

**Soundness:** 3
**Presentation:** 3
**Contribution:** 3
**Rating:** 7
**Confidence:** 1

**Summary:**

- This paper introduces a method using an online algorithm with class priority reweighting to downsample data for vision and text tasks. The experiments demonstrate that this approach can achieve robustness and efficiency in situations with imbalanced classes and poor worst-class generation performance.

**Strengths:**

- Originality and Significance: The new online batch selection approach is robust to noise, imbalance, and distributional shifts. It can effectively preserve the worst-class generation performance, as demonstrated by the results.
- Quality and Clarity: The work includes positive experimental results supported by sufficient formulas that recap the background, problem definition, theoretical algorithms, and justification of the method. The use of figures helps readers better understand the algorithm and evaluation results.

**Weaknesses:**

- The ablation studies are not elaborated upon. There are limited descriptions on how to draw conclusions about the necessity of each component for robust online batch selection, which raises curiosity.

**Questions:**

- What are the most challenging aspects when applying this new method to scaling and broader scenarios? Are there potential solutions for addressing these challenges?

**Limitations:**

- Further exploration is needed on the balance between compute efficiency and data downsampling efficiency among different combinations of model architectures and class group numbers.
- Limited tasks: The work focuses on and is evaluated using a series of text and image classification tasks. However, to improve data efficiency for real-world applications and broader machine learning models, further research on the effectiveness of this method on other tasks is necessary.

---

> ### Author Rebuttal · Authors · 2024-08-07
>
> We thank the Reviewer for their score and recognising the originality and significance of our approach along with the quality and clarity of our manuscript. In response to the weaknesses, question, and limitations:
>
> > W: Ablation Studies
>
> In Section 5.2 we investigate how the removal of the class holdout loss term affects the performance of the algorithm. Later in Appendix A.7.2 we also describe in detail how the clipping of the selection score stabilises the selection scores and thus motivate its inclusion in the algorithm. In response to the reviewers feedback we will look to include the following in the paper to further motivate the inclusion of the model loss and class irreducible loss model terms in the algorithm.
>
> *“In Figure 4a) we note that removing the Model Loss results in the worst performance in the set of ablation studies. This is because the Model Loss provides REDUCR with information about which points are currently not classified correctly by the model. By removing this term REDUCR only selects points which do well under the Class Irreducible Loss model and does not prioritise points the model has not yet learnt. Selecting points not yet learnt by the model is an important quality in online batch selection approaches and the main premise of the Train Loss baseline algorithm.*
>
> *Likewise by removing the Class Irreducible Loss Model term we remove the ability of the model to infer if a point can be learnt or not. In [1], the authors note that these pretrained models enable the algorithm to pick points that are learnable and do not have label noise.“*
>
> > Q: Challenges
>
> The most challenging aspect, which we wrote in Section 6 (Limitations), is that the computational efficiency of REDUCR scales linearly with the number of classes and as such applying this method to scenarios that scale with the number of classes increases the computational cost of selecting data with REDUCR.
>
> To begin addressing these problems we explore an approach in which we apply REDUCR to superclasses in Section 5.3. This approach shows strong results on the CIFAR100 dataset, outperforming the worst-class and average test accuracy of the baseline algorithms. On top of this we propose another direction of research in the limitations section - using smaller models to guide larger models. A variety of work (see [1][2] in our related work) has shown that smaller models can guide the training of a larger model and, as such, we think this is a promising direction of investigation for improving the algorithm’s computational complexity.
>
> We intend to study this more rigorously in future work as the scope is beyond that of one conference paper.
>
> > L1: Generalisation of REDUCR
>
> The compute efficiency and data downsampling efficiency are reflected in the progression of test accuracy as the number of training steps increases. In the general response, we include results on the ConvNext model architecture applied to the Clothing1M dataset. The results indicate that REDUCR can be widely applicable to more architectures. We agree with the reviewer that improving compute efficiency for a larger number of classes is a promising direction of research and address this in the Limitations section of the paper.
>
> >L2: Effectiveness of REDUCR
>
> We respectfully disagree with the reviewer on this point. Whilst further research on applying REDUCR to non-classification tasks is a promising direction for future work, the focus of this paper is on data selection in the classification setting, introducing the robust online batch selection problem and proposing a practical solution. We demonstrate REDUCR’s performance on two different modalities of data (text and image) showing the algorithm generalises across a number of different real-world classification tasks. Classification problems are still highly relevant to practitioners (e.g. sentiment classification to ensure LLM output is safe) and as such REDUCR has numerous real-world use cases as it is described in our work.
>
> While it is always nice to include more tasks and more applications, we don’t consider those “necessary” for our paper, since we have already demonstrated a range of detailed results in both the main paper and appendix to support our claim that class-robust data downsampling is an important problem and REDUCR can help to solve this problem.
>
> ***References***
>
> [1] Mindermann et al. Prioritized training on points that are learnable, worth learning, and not yet learnt. International Conference on Machine Learning, 2022.
>
> [2] Sang Michael Xie et al. Doremi: Optimizing data mixtures speeds up language model pretraining. arXiv preprint arXiv:2305.10429, 2023.

---

> > ### Comment · Reviewer_9L5X · 2024-08-08
> >
> > Thanks for the comprehensive responses, especially regarding the challenges. I have updated the rating to 7 and look forward to the future work.

---

> > > ### Author Response · Authors · 2024-08-12
> > >
> > > We would like to thank the reviewer for their remarks and raising their score.

---

### Official Review · Reviewer_L9Bs · 2024-07-13

**Soundness:** 3
**Presentation:** 4
**Contribution:** 3
**Rating:** 7
**Confidence:** 3

**Summary:**

**Problem**: The paper addresses the challenge of efficient online batch selection while training for classification tasks. It identifies a key issue with current batch selection algorithms: they often perform poorly on underrepresented classes in the training data. The paper proposes solutions to this problem while ensuring efficient data sampling. The goal is to optimize both the worst class performance and the average class performance on classification benchmarks by selecting a subset of each batch that enhances the performance of the worst performing class.

**Contributions**: The paper proposes an online batch selection algorithm that assigns priority weights to training data points in a batch based on i) class weights in training data and ii) usefulness to the model. This is done by selecting points which have the highest worst class generalization error (hence selecting underrepresented classes) and selecting those points which are learnable given more data, thus avoiding noisy and task-irrelevant points. All of these objectives are ensured by assigning selection scores to training points in a batch. These selection scores are computed based on the model loss (training loss), class-irreducible loss (training loss from a reference model training on class-specific holdout data), and class-holdout loss (training loss on holdout data). Although existing works such as RHO-LOSS employ similar reference models training on holdout datasets, they fail to optimize for underrepresented classes in the training data. Further, REDUCR is also robust to noise in the training data, by selecting points that are harder to learn from worse-performing classes over those which are easier to learn (and are already performing well at a given training point).

**Experiments and Results**: REDUCR has been evaluated on both image and text classification datasets. The paper assesses both worst-class test accuracy and average test accuracy. REDUCR outperforms existing batch selection methods like RHO-LOSS by at least 3% in worst-class accuracy, while achieving comparable results to RHO-LOSS in average class performance.

**Strengths:**

The strengths of the paper are:

1. The proposed data sampling algorithm, REDUCR, aims to optimize worst-class performance in classification tasks. REDUCR makes significant progress towards efficient data sampling algorithms that are robust to class imbalance and noise in training data.
2. Contributions regarding improved worst-class accuracy are empirically well-supported across a wide range of classification datasets, including both text and images.
3. The REDUCR algorithm is well-motivated in the paper, with a focus on the efficiency of data sampling.

**Weaknesses:**

Weaknesses of the paper:

1. All experiments are conducted on small-scale datasets (with fewer than 1000 classes). The paper mentions in line 73 that the problem of robustness becomes less applicable in settings with a high number of classes. However, as the number of classes increases, the problem of class imbalance becomes more pronounced due to the power law in most datasets. Since REDUCR scales linearly with the number of classes, a thorough evaluation of the suggested approach of using super-classes would be beneficial.
2. The paper does not compare with other approaches, such as [1], which specifically address popularity bias in training datasets and could improve worst-class performance.
3. The paper lacks ablation experiments on hyperparameters in REDUCR, such as the fraction of data points selected for training and the number of reference models trained.


[1] [Distributionally Robust Language Modeling](https://arxiv.org/pdf/1909.02060)

**Questions:**

Broadly two questions on REDUCR evaluation:

1. How does REDUCR compare with existing data selection algorithms (other than RHO-LOSS) aimed specifically at distributional shifts?
2. How does REDUCR perform on large-scale classification tasks? Further, can REDUCR be applied to tasks other than classification?

**Limitations:**

Please refer weaknesses and questions.

---

> ### Author Rebuttal · Authors · 2024-08-07
>
> We thank the reviewer for their score recognising the strengths of our paper, including the strong motivation and progress towards solving the problem setting we introduce in this work. In response to the weaknesses and questions:
>
> > W3: Ablation Experiments
>
> We direct the reviewer to Appendix A.7 of our work where we run ablations on the gradient weight, learning rate and percent train hyperparameters. In response to this feedback we will make it clearer in Section 5.2 (Ablation Studies) that these hyperparameters studies are in the paper’s appendix. The number of reference models is the same as the number of classes if we don’t adopt superclasses. We discuss the superclass setting in the response to question “The paper mentions in line 73..”
>
> > W2/Q1: Comparison with Existing Algorithms
>
> To the best of our knowledge we are the first work to consider the setting of **online batch selection** that is robust to issues such as distributional shifts and class imbalance. Whilst approaches such as [1] and [2] derive training objectives to address distributional shift, these approaches do not address the question of how to sub-sample the data they are trained on. REDUCR selects data and thus adjusts the distribution of classes in the loss, however [2] directly upweights the gradients of poorly performing classes in the training loss. Indeed one could apply this approach on top of REDUCR to try and further improve performance on the worst classes. This is an important direction to explore but orthogonal to the approach we introduce in REDUCR.
>
> > W1/Q2: High Number of Classes
>
> To address the linear scaling problem, we propose the super-class approach in Section 5.3 and apply it to CIFAR100. We note that the results on CIFAR100 with super-classes are promising, CIFAR100’s worst-class average test accuracy and average test accuracy outperform all the baseline algorithms. We note in the limitations that this is one of many potential approaches to address the linear scaling problem and as such a full investigation of all these ideas is left for future work; our goal in this work is to introduce the problem setting and thoroughly investigate the base REDUCR algorithm.
>
> The reviewer also addresses the problem of class imbalance becoming more pronounced due to the power law. This requires further investigation beyond the scope of our work to answer important questions e.g. what are the real world scenarios where introducing more classes worsens class imbalance, what datasets are representative of this, and how these questions are reflected in the model’s performance.
>
> > Q2: General Applicability of REDUCR
>
> REDUCR can potentially be applied to generative tasks or regression. One can generalise the idea of optimising the worst-class performance to optimising the worst-group performance [2]. For example, generative tasks can have different groups of problems related to math, coding, or literature etc. Regression tasks may have different groups of inputs. To apply REDUCR in this setting one needs group labels for each data point and an objective (e.g., log-likelihood) such that the REDUCR selection rule (Equation 7) can still be applied. REDUCR lays a strong foundation for future exploration in this direction, specifically in Section 5.3 where we group classes together, and show strong empirical performance on the CIFAR100 dataset with this approach.
>
> ***References***
>
> [1] Yonatan Oren, Shiori Sagawa, Tatsunori B Hashimoto, and Percy Liang. Distributionally robust language modeling. arXiv preprint arXiv:1909.02060, 2019.
>
> [2] Shiori Sagawa et al. Distributionally robust neural networks for group shifts: On the importance of regularization for worst-case generalization. Internation Conference on Learning Representations, 2020.

---

### Author Rebuttal · Authors · 2024-08-07

We thank the reviewers and ACs for their time and work in evaluating our paper. In response to your feedback we have re-run our experiments on the Clothing1M dataset with a new model architecture to test the generalizability of REDUCR. The results can be seen in the attached PDF. REDUCR outperforms the next best baseline in terms of the mean worst-class and average test accuracy. We ran each experiment for 3 seeds, using the original hyperparameters from our original Clothing1M experiment. These are promising initial results and further hyperparameter tuning specific to the model architecture will improve the performance further.

We have also clarified several points in the manuscript. Firstly we have provided a more detailed explanation of how ablating various components of REDUCR affects the algorithm’s performance:

*“In Figure 4a) we note that removing the Model Loss results in the worst performance in the set of ablation studies. This is because the Model Loss provides REDUCR with information about which points are currently not classified correctly by the model. By removing this term REDUCR only selects points which do well under the Class Irreducible Loss model and does not prioritise points the model has not yet learnt. Selecting points not yet learnt by the model is an important quality in online batch selection approaches and the main premise of the Train Loss baseline algorithm.*

*Likewise by removing the Class Irreducible Loss Model term we remove the ability of the model to infer if a point can be learnt or not. In [1], the authors note that these pretrained models enable the algorithm to pick points that are learnable and do not have label noise.“*

Secondly, we have clarified how by analysing the worst-class test accuracy we lower bound the performance of the trained model under a variety of class distribution shifts:

*“Finally, it is important to note that we analyse the worst-class test accuracy which can be interpreted as a lower bound on a model’s performance under all class distribution shifts. This is because the worst possible distribution shift between the training and test set is one where the entire test set consists of only points from the worst performing class.”*

Thirdly we have visualised the distribution shift in the Clothing1M dataset, we will include this in the paper and have added this plot to the attached PDF.

Finally, we will further clarify in the main text that the extensive ablation experiments we ran on the gradient weight, learning rate and percent train hyperparameters are included in the Appendix.

We look forward to engaging with the reviewers in the author reviewer discussion period.

---

### Decision · Program_Chairs · 2024-09-25

**Decision:**

Accept (poster)

**Comment:**

This paper investigates the problem of reducing model training costs. The authors propose a novel approach that combines an online algorithm with class priority reweighting to downsample data effectively. Experiments conducted on both image and text datasets demonstrate the efficacy of the proposed method. Overall, this paper is technically novel and solid, clearly exceeding the bar for NeurIPS. All reviewers agree to accept it for publication. Please revise this paper according to the reviewers' suggestions in the final version.